# Multivariate Time Series Analysis of Temperatures in the Archaeological Museum of L’Almoina (Valencia, Spain)

**DOI:** 10.3390/s21134377

**Published:** 2021-06-26

**Authors:** Sandra Ramírez, Manuel Zarzo, Fernando-Juan García-Diego

**Affiliations:** 1Department of Applied Statistics, Operations Research and Quality, Universitat Politècnica de València, Camino de Vera s/n, 46022 Valencia, Spain; smramirez@javerianacali.edu.co (S.R.); mazarcas@eio.upv.es (M.Z.); 2Department of Natural Sciences and Mathematics, Pontificia Universidad Javeriana Cali, Cali 760031, Colombia; 3Department of Applied Physics (U.D. Industrial Engineering), Universitat Politècnica de València, Camino de Vera s/n, 46022 Valencia, Spain

**Keywords:** ARIMA, art conservation, Holt–Winters, k-means, random forest, sensor diagnosis, sPLS-DA

## Abstract

An earlier study carried out in 2010 at the archaeological site of L’Almoina (Valencia, Spain) found marked daily fluctuations of temperature, especially in summer. Such pronounced gradient is due to the design of the museum, which includes a skylight as a ceiling, covering part of the remains in the museum. In this study, it was found that the thermal conditions are not homogeneous and vary at different points of the museum and along the year. According to the European Standard EN10829, it is necessary to define a plan for long-term monitoring, elaboration and study of the microclimatic data, in order to preserve the artifacts. With the aforementioned goal of extending the study and offering a tool to monitor the microclimate, a new statistical methodology is proposed. For this propose, during one year (October 2019–October 2020), a set of 27 data-loggers was installed, aimed at recording the temperature inside the museum. By applying principal component analysis and k-means, three different microclimates were established. In order to characterize the differences among the three zones, two statistical techniques were put forward. Firstly, Sparse Partial Least Squares Discriminant Analysis (sPLS-DA) was applied to a set of 671 variables extracted from the time series. The second approach consisted of using a random forest algorithm, based on the same functions and variables employed by the first methodology. Both approaches allowed the identification of the main variables that best explain the differences between zones. According to the results, it is possible to establish a representative subset of sensors recommended for the long-term monitoring of temperatures at the museum. The statistical approach proposed here is very effective for discriminant time series analysis and for explaining the differences in microclimate when a net of sensors is installed in historical buildings or museums.

## 1. Introduction

The environmental conditions of historical buildings, exhibition facilities and storage areas in museums have been shown to be the most crucial factor in the preservation of collections and artifacts. Temperature, humidity and lighting can potentially deteriorate or even destroy historical or cultural objects that are kept, protected and displayed in collections [1]. A continuous monitoring of the indoor environment can provide information about the microclimatic conditions affecting the works of art. Monitoring is an essential tool for developing a preventive control program aimed at maintaining the optimal microclimatic conditions for preservation. As a consequence, long-term monitoring has to be applied to prevent the deterioration of artworks [2]. Furthermore, it is necessary to find practical solutions and tools for the incorporation of climate change adaptation in the preservation and management of cultural heritage [3]. In particular, in archaeological sites, temperature differences between various minerals in block surfaces and alternative surfaces cause thermal stress. Humidity and thermal stresses are important causes of microfractures between the mineral grains of blocks [2].

In L’Almoina museum (Valencia, Spain), a pronounced gradient of temperature was found [4] due to a skylight which was included in the architectonical design of the museum (Figure 1). A thermo-hygrometric monitoring study carried out in this museum in 2010 [4] discussed the significant effect of the skylight on the variations in T and RH. A pronounced greenhouse effect was noted, as a consequence of the skylight and the high temperatures reached in summer in Valencia. In 2011 the layer of water was removed due to a leak that had to be repaired. To replace the beneficial effect of the water layer, a provisional canvas cover was installed directly over the skylight, in order to avoid the overheating of the archaeological site below, by preventing direct sunlight. A second thermo-hygrometric monitoring study was performed in 2011 to assess the effect of different corrective measures and changes implemented in the museum [5]. The microclimatic data of RH and T recorded in 2010 before laying the canvas cover was compared with air conditions in 2013, after its installation. It was found that the presence of the canvas covering the skylight improved the T and RH conditions, so that the microclimate was in accordance with the international standards [6,7].

Given the marked detrimental influence of the skylight, a long-term monitoring is required for the control of thermal conditions. Thus, it is necessary to find practical solutions and tools for the preservation and management of the ruins. For this purpose, a statistical methodology for classifying different time series of temperature that are very similar is of interest. Such methodology can help to characterize different zones in the museum and to provide guidelines for monitoring the thermal conditions.

Some studies have been carried out in order to propose a plan for monitoring either temperature (T) or relative humidity (RH) for art conservation. Three reported studies proposed a methodology for classifying different time series using either observations of time series, or features from time series. García-Diego and Zarzo [8] and Zarzo et al. [9] applied Principal Component Analysis (PCA) in order to study the values of RH from sensors installed in different positions at the apse vault of Valencia’s Cathedral (Spain). Zarzo et al. [9] reported that the first and second principal components could be estimated according to a linear combination of the average RH values and the moving range of RH. Based on the two first components, the differences between the time series of RH in different positions in the apse were discussed. Ramírez et al. [10] proposed a statistical methodology in order to classify different time series of RH, which pointed to those zones with moisture problems in the apse vault of the Cathedral of Valencia. Merello et al. [11] analyzed 26 different time series of RH and T, recorded at Ariadne’s house (Pompeii, Italy), using graphical descriptive methods and Analysis of Variance (ANOVA), in order to assess the risks for long-term conservation of the mural paintings in the house. The work provided guidelines about the type, calibration, number and position of thermo-hygrometric sensors in outdoor or semi-confined environments.

The proposed methodology in this article has an adequate capability for discriminating time series with similar features. In addition, it can help to obtain parsimonious models with a small subset of variables leading to a satisfactory discrimination. As a consequence, its results can be easily interpreted and can help to select a subset of representative sensors for the long-term monitoring of indoor air conditions inside the museum. Finally, this methodology can be effective in order to establish the different zones in the archaeological site and to discriminate the microclimate of these areas.

**Figure 1 sensors-21-04377-f001:**
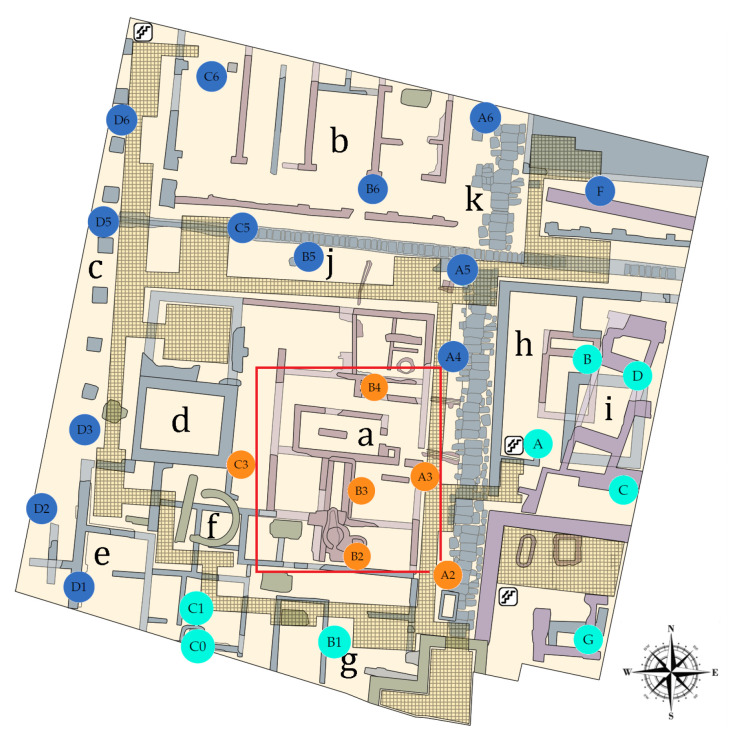
Plan of the L’Almoina archaeological site, indicating the position of 27 data-loggers for monitoring the air conditions inside the museum. Based on the multivariate analysis of temperatures (see Section 3.3.6), three zones were established: North West (NW, in blue), South East (SE, in green) and Skylight (Sk, in orange). The different observable structures and the construction phases in the museum are indicated: (a) Roman baths; (b) Imperial granary; (c) Portico of the imperial forum; (d) Imperial chapel; (e) Imperial basilica; (f) Byzantine apse and tombs; (g) Byzantine Cathedral Baptistery; (h) Republican and Imperial Asklepieion; (i) Alcázar Aldalusí; (j) Decumani; and (k) Cardus [12].

Aimed at better understanding the differences of microclimate in L’Almoina archaeological site (Figure 2), a set of 27 autonomous data-loggers was installed at different points of the museum (Figure 1). The time period under study was of about one year, from 22 October 2019 to 20 October 2020. The main goal of this research was to identify different microclimates at the museum and to characterize the differences in temperature between such zones. The purpose was to classify the sensors according to features and variables extracted from the time series of T. Another target was to identify those variables that best discriminate the different time series per zone. For this purpose, a methodology was applied based on sparse partial least squares discriminant analysis (sPLS-DA) and random forest (RF) with models and functions of time series [10,13]. Another target was to identify a subset of representative sensors for a long-term monitoring of thermic air conditions in the museum, as well as to determine the best location recommended for these sensors. The proposed methodology is rather new in the context of clustering of time series applied to cultural heritage. Furthermore, this methodology can be useful for defining different zones in the museum, according to features of the time series of T, as well as to achieve a correct classification of all sensors in such zones.

This article is structured as follows. In Section 2, a short background related to art conservation is presented. Characteristics of the dataset and the sensors, criteria for determining the stages of the time series of T, methods for calculating features of the time series and strategies for classifying time series are introduced in Section 3. The most notable results of the different analyses and their discussion are presented in Section 4. Finally, conclusions appear in Section 5.

## 2. Background

### 2.1. Studies for the Long-Term Preservation of Artworks

Many studies have been conducted in recent years which monitor the climatic parameters for the long-term preservation of cultural heritage. Frasca et al. [14] studied the impact of T, RH and carbon dioxide (CO${}_{2}$) on the organic and hygroscopic artworks in the church of Mogiła Abbey. They found that artworks were at high risk of mechanical damage for approximately 15% of the time under study, due to an excessive variability of RH. Huijbregts et al. [15] proposed a method for evaluating the damage risk of long-term climate changes on artifacts in museums or historic buildings. This method was applied for two historic museums in the Netherlands and Belgium. For the examined case studies, they found that the expected climate change would significantly increase both indoor T and RH, with the increase of the latter having the highest impact on the damage potential for artifacts in museums. Zítek and Vyhlídal [16] proposed a novel air humidity control technique for preventing the moisture sensitive materials from varying the equilibrium of their moisture content, maintaining desirable environmental conditions for the preventive conservation of cultural heritage. Angelini et al. [17] designed and installed a wireless network of sensors for monitoring T and RH, aimed at establishing a correlation between the environmental conditions and the conservation state of artifacts. In addition, Lourenço et al. [18] studied air T and RH, among other parameters, in the historical city center of Bragaça (Portugal). Other researchers have studied the variation of some environmental parameters, in order to identify the main factors involved in the deterioration of certain remains, e.g., exposed and buried remains at the fourth century Roman villa of Chedworth in Gloucester, England [19].

### 2.2. European Standards

Experts suggest that it is necessary to investigate the actual environmental dynamics in a museum before any structural intervention. Furthermore, it is important to define the compatibility between the climate control potentials and the preservation requirements [2]. Several European Standards [20,21,22,23,24,25,26] have been developed for the monitoring, elaboration and study of the microclimatic data, as supporting actions for the preservation of artifacts. Long-term monitoring is required, as well as an appropriate statistical approach for the data management.

A large economical investment is being provided by governments within the European Union to preserve artworks in museums. Different research projects have monitored the indoor microclimate within museums, in order to analyze the relationship between thermo-hygrometric conditions and the degradation of materials, from which works of art are made. For example, with the goal of preserving artwork and artifacts, the CollectionCare Project is currently working on the development of an innovative system for making decisions about the preventive conservation of artworks in small- to medium-sized museums and private collections [27].

### 2.3. Characteristics of the L’Almoina Museum

The archaeological site of L’Almoina in Valencia (Spain) is an underground museum at about 3 m below the city floor level. It occupies an area of about 2500 $m^{2}$. The archaeological remains are covered by a concrete structure, which forms an elevated plaza above the ruins. This cover connects with walkways and steps at different heights along its perimeter. There is no vertical retaining wall inside the museum to isolate the remains and prevent water diffusion through capillarity from the surrounding areas. An external glass skylight (225 $m^{2}$) was adapted to the museum so that part of the ruins could be observed from the pedestrian plaza. Nowadays, the skylight is protected by a layer of water (see Figure 2) to prevent high temperatures of the glass.

## 3. Materials and Methods

### 3.1. Materials: Description of the Datasets

In total, 27 data-loggers were installed for the monitoring of T and RH at L’Almoina museum. Technical details are as follows: data-logger model WS-DS102-1, with a temperature range between −40 and $+60{\;}^{\circ }C$ and an accuracy according to the MI-SOL manufacturer of $\pm 1{\;}^{\circ }C$, under 0–50${\;}^{\circ }C$ [28]. The data acquisition rate was of one recording every five minutes, so that a manual download of data was necessary every two months. The monitoring experiment started on 22 October 2019 and ended on 20 October 2020. The initial number of available observations of T was approximately 104,832 per sensor (i.e., 364 days·24 h/day·12 values/h), which were arranged as a matrix containing 27 columns (i.e., temperatures recorded by each sensor) by 104,832 rows. Daily cycles are clearly marked, which implies a repetitive pattern every 288 values, but such a data sequence seems too long. Thus, it was decided to calculate the median of values recorded per hour, which leads to daily cycles every 24 values. This frequency seems more convenient for the use of seasonal methods of time series analysis. Thus, a new matrix was arranged comprising 8730 observations by 27 sensors. This dataset did not contain missing values.

### 3.2. Data Calibration

All sensors were calibrated prior to their installation by means of an experiment carried out inside a climatic chamber, model alpha 990-40H from Design Environmental Ltd. (Gwent, UK). The temperature was maintained at three different levels: 5, 23 and $30{\;}^{\circ }C$. For each stage, the RH was 75%, 50% and 30%, respectively. Each stage of temperature was maintained for 2 h, so that the total calibration experiment lasted for 6 h. The frequency of temperature recorded was one datum per five minutes from each sensor. Next, the median of T per hour was calculated by sensor. Linear regression was applied to obtain calibration functions, one per sensor, relating the measured median of T as a function of the real temperature inside the chamber. Finally, the temperature matrix (8730×27) was modified by correcting the bias of each sensor, according to the resulting calibration functions, which leads to a corrected matrix containing the median temperature per hour, after the calibration.

### 3.3. Statistical Methods

The statistical methodology comprises the following steps: (1) Identify structural breaks in all time series. (2) Extract features directly from the time series. (3) Compute classification variables using the additive seasonal Holt–Winters approach. (4) Compute classification variables by means of seasonal ARIMA. (5) Compute classification variables according to the Wold decomposition. (6) Determine the optimum number of classes and establish the class for each sensor, using PCA and k-means algorithm. (7) Check the classification of sensors using sPLS-DA and identify the optimal subset of variables that best discriminate between classes, per method. (8) Check the classification of sensors by means of the RF algorithm and identify the optimal set of variables calculated from each method, that best discriminate between the classes. (9) Propose a methodology for selecting a subset of representative sensors for future long-term monitoring experiments in the museum.

The main R software packages [29] (version 4.3) used to carry out the statistical analyses were: mixOmics [30,31], aTSA [32], forecast [33,34], strucchange [35], tseries [36], moments [37], PerformanceAnalytics [38], NbClust [39] and QuantTools [40].

#### 3.3.1. Identification of Structural Breaks in the Time Series

In real conditions, time series can undergo sudden shifts or changes in the slope of a linear trend. Such events are known as structural breaks [41]. The CUSUM and supF tests, among others, can be used to detect structural breaks in a time series [42,43]. By carefully inspecting the evolution of all observed time series of temperature (T) over time (Figure 3), certain potential structural breaks can be observed. Both the CUSUM and supF tests were applied after computing the logarithmic transformation and one regular differentiation to the time series. Such logarithmic transformation was intended to stabilize the variance, while the regular differentiation was used to remove the trend of the different time series [44]. The notation used throughout this paper is the following: r refers to the logarithmic transformation of T and W denotes one regular differentiation of the logarithmic transformation of T. Thus, each value of W corresponds to $w_{t}=r_{t}-r_{t-1}$, being $r_{t}=\ln\left(T_{t}\right),t=1,\cdots ,t_{max}$. The tests were computed with functions Fstats and efp from the strucchange package [35]. Initially, 5 stages were tentatively established: warm 1 (Wr1, comprising n = 1490 observations), cold (Cd, n = 1703), transition (Tr, n = 2303), hot (Ht, n = 2327) and warm 2 (Wr2, n = 903) (see Figure 3). Wr1 corresponds to 22 October–23 December 2019, Cd to 24 December–3 March 2020, Tr to 4 March–7 June, Ht to 8 June–12 September and Wr2 to 13 September–20 October 2020.

#### 3.3.2. Calculation of Classification Variables—Method M1

This method consists of computing features such as mean, median and maximum, among others, from estimates of the Auto Correlation Function (ACF), Partial Auto Correlation Function (PACF), spectral density and Moving Range (MR) [45,46]. The ACF and PACF correlograms of the observed time series are commonly used to fit Auto Regressive Moving Average (ARMA) models. Firstly, a set of variables denoted as Type 1 comprised the mean, MR and PACF, which were estimated for the 8 stages of T values. The average of T was calculated to capture the level or position in each stage of the time series. The mean of MR with order 2 (i.e., average range over 2 past values) was computed to identify sudden shifts or increases in the level of T. For each stage of T, the sample PACF parameters ($\alpha_{l}$ at lag *l*) were estimated for the first four lags (*l* = 1, 2, 3, 4), which are usually the most important ones for capturing the relevant information in time series.

Secondly, another set of variables called Type 2 comprised spectral density and ACF, which were estimated for T values after applying the logarithm transformation and regular differencing. The objective of using this transformation and differencing was to stabilize the variances and remove the trend of T, so that the computed variables provide information about the seasonal component of the time series. Spectral density was estimated using the periodogram of observed time series W (I(w) of signals *w*). The maximum peak of the periodogram and its frequency were identified. Values of ACF ($\rho_{l}$ at lag *l* ) were estimated to analyze the correlation between W values with the lagged values of the same observed time series at each lag, for the first 72 lags. This criterion was used because the values of the ACF correlogram for further lags were comprised within the limits of a 95% confidence interval in the correlogram.

The different steps involved in M1 are depicted in Figure 4. Firstly, the 27 time series were split according to the climatic stages observed: Wr1, Cd, Tr, Ht and Wr2 (Data 2). Secondly, some of the main stages (Data 2) were subdivided according to the structural breaks (SB) identified in Wr1, Cd and Ht (Data 3). In the third step (Data 4), the logarithm transformation was applied and, next, one regular differentiation (Data 5). The fifth step consists of applying the formulas of Type 2 variables to $w_{t}$: maximum of periodogram ($M_{I(w)}$) and its frequency (*w*), as well as mean, median, range and variance of the sample ACF for the first 72 lags ($\mu_{{\hat{\rho }}_{l}}$, $Md_{{\hat{\rho }}_{l}}$, $R_{{\hat{\rho }}_{l}}$ and $\sigma_{{\hat{\rho }}_{l}}^{2}$, with l=1,⋯,72). Finally, the formulas of Type 1 variables were applied to T values (Data 3): mean of T ($\mu_{T}$), mean of MR of order 2 ($\mu_{MR}$) and PACF for the first four lags ($\alpha_{1}$, $\alpha_{2}$, $\alpha_{3}$ and $\alpha_{4}$).

The estimates of MR values were computed with the R software according to the function rollrange from the QuantTools package [40]. The sample ACF and sample PACF values were calculated with the function acf (stats) [47] and pacf (tseries) [36], respectively. The values of the periodogram and their frequencies were obtained with the function spectrum (stats).

#### 3.3.3. Calculation of Classification Variables—Method M2

The Holt–Winters method (H-W) [48] is a type of exponential smoothing that is used for forecasting in time series analysis. The seasonal H-W (SH-W) method uses a smoothing equation for each component of a given time series: the level, slope and seasonality, which are denoted at time $t^{*}$ as $a_{t^{*}}$, $b_{t^{*}}$ and $s_{t^{*}}$, respectively. The additive SH-W prediction function of a time series of T is given by Equation (Equation 1), where *p* denotes the number of observations per period and *k* is the integer part of (l−1)/p [49,50]. This equation was implemented with the conditions: 0≤α≤1, 0≤β≤1, 0≤γ≤1 and $t^{*}>s$. When the algorithm converges, *a* corresponds to the last level of $a_{t^{*}}$, *b* is the last slope of $b_{t^{*}}$ and $s_{1}$–$s_{24}$ are the last seasonal contributions of each $s_{t^{*}}$. A forecast of ${\hat{t}}_{t^{*}+l}$ based on all the data up to time $t^{*}$ is denoted by ${\hat{t}}_{t^{*}+l|t^{*}}$; a simpler notation for ${\hat{t}}_{t^{*}+l|t^{*}}$ is ${\hat{t}}_{t^{*}+l}$.
(1)$$\begin{array}{cc} {\hat{t}}_{t^{*}+l|t^{*}} & =a_{t^{*}}+lb_{t^{*}}+s_{t^{*}+l-p\left(k+1\right)},\mathrm{where}\\ a_{t^{*}} & =\alpha \left(t_{t^{*}}-s_{t^{*}-p}\right)+\left(1-\alpha \right)\left(a_{t^{*}-1}+b_{t^{*}-1}\right)\\ b_{t^{*}} & =\beta \left(a_{t^{*}}-a_{t^{*}-1}\right)+\left(1-\beta \right)b_{t^{*}-1}\\ s_{t^{*}} & =\gamma \left(t_{t^{*}}-a_{t^{*}-1}-b_{t^{*}-1}\right)+\left(1-\gamma \right)s_{t^{*}-p} \end{array}$$

The method M2 consists of fitting additive SH-W equations in two steps. Firstly, for each stage of the 27 time series of T, the classification variables are the last level of smoothing components of the additive SH-W method per sensor: level (*a*), slope (*b*) and seasonal components ($s_{1},s_{2},\ldots ,s_{24}$). The method was fitted by considering p=24 as the number of observations per day. These smoothing components were called as Type 3 variables. Secondly, by considering the complete time series, the first 24 predictions of T for each unique additive SH-W model per sensor were regarded as additional classification variables, which were denoted as Type 5 variables.

Although a residual analysis is not necessary when using SH-W method, estimates of the features from residuals were also computed per stage of the time series (Type 4 variables): sum of squared estimate of errors (SSE), maximum of periodogram ($M_{I(w)}$) and its frequency (*w*) and several parameters (mean, median, range and variance) of sample ACF for 72 lags ($\mu_{{\hat{\rho }}_{l}}$, $Md_{{\hat{\rho }}_{l}}$, $R_{{\hat{\rho }}_{l}}$ and $\sigma_{{\hat{\rho }}_{l}}^{2}$, with l=1,⋯,72). Moreover, the Kolmogorov–Smirnov (KS) normality test [51] and Shapiro–Wilk test (SW) [52,53,54] were applied in order to extract further information from the different stages of the observed time series. The statistic of the KS test (denoted as Dn) was used to compare the empirical distribution function of the residuals with the cumulative distribution function of the normal model. Likewise, the statistic of the SW test (Wn) was employed to detect deviations from normality, due to skewness and/or kurtosis. The statistics of both tests were also used as classification variables because they provide information about deviation from normality for the residuals derived from the SH-W method.

The steps involved in M2 are depicted in Figure 5. Firstly, the different time series were split according to the climatic stages observed (Data 2) and the structural breaks (SB) identified (Data 3). Secondly, the seasonal H-W method was applied to Data 3 in order to obtain the last level of smoothing components (Type 3 variables) and then the model residuals. The third step consisted of applying the formulas of Type 4 variables to the residuals. Finally, the seasonal H-W method was applied to Data 1 in order to obtain the first 24 predictions of T (Type 5 variables).

The HoltWinters function (stats) was used to fit the Additive SH-W method. The shapiro.test (stats) and ks.test (dgof) [55] were used to apply the normality tests. Values of the sample ACF and sample PACF were computed with the functions acf (stats) and pacf (tseries), respectively. Values of the periodogram and their frequencies were calculated with the function spectrum (stats).

#### 3.3.4. Calculation of Classification Variables—Method M3

The ARMA model is also known as the Box–Jenkins approach, which focuses on the conditional mean of the time series and assumes that it is stationary [44]. By contrast, the ARIMA model can be used when a time series is not stationary. It employs regular differencing of the time series prior to fitting an ARMA model. ARIMA (p,d,q) models employ *p* Auto Regressive (AR) terms, *d* regular differences and *q* Moving Average (MA) terms. Parameters of the AR component are denoted as $\phi_{i}$ (i=1,⋯,p) and parameters of the MA component as $\theta_{j}$ (j=1,⋯,q). The error terms $\epsilon_{t}$ are assumed to be a sequence of data not autocorrelated with a null mean, which is called White Noise (WN) [44]. In addition, if a given time series is assumed to follow an ARIMA process, the conditional variance of residuals is supposed to be constant. If this is not the case, then it is assumed that an ARCH effect exists in the time series. Two of the most important models for capturing such changing conditional variance are the ARCH and Generalized ARCH (GARCH) models [41].

Seasonal ARIMA $\left(p,d,q\right){\left(P,D,Q\right)}_{S}$ models are more appropriate in this case given the marked daily cycles. *P* refers to the number of seasonal AR (SAR) terms, *D* to the number of difference necessary to obtain a stationary time series, *Q* to the number of seasonal MA (SMA) terms and *S* to the number of observations per period (S=24 in this case). Parameters of the SAR component are denoted as $\Phi_{i}$ (i=1,⋯,P) and the SMA component as $\Theta_{j}$ (j=1,⋯,Q). The error terms $\epsilon_{t}$ are assumed to be a WN sequence [44].

A seasonal ARIMA $\left(p,d,q\right){\left(P,D,Q\right)}_{S}$ model is given by Equation (Equation 2), where the polynomial $\phi_{p}\left(B\right)$ is the regular AR operator of order *p*, $\theta_{q}\left(B\right)$ is the regular MA operator of order *q*, $\Phi_{P}\left(B^{S}\right)$ is the seasonal AR operator (SAR) of order *P*, $\Theta_{Q}\left(B^{S}\right)$ is the seasonal MA operator (SMA) of order *Q* and *B* is the backshift operator (i.e., $Bv_{t}=v_{t-1}$, for t>1 and $B^{24}v_{t}=v_{t-24}$, for t>24). Furthermore, $\nabla_{S}^{D}$ represents the seasonal differences while $\nabla^{d}$ accounts for the regular differences, so that $\nabla_{S}^{D}$ is defined as ${\left(1-B^{S}\right)}^{D}$ and $\nabla^{d}$ as ${\left(1-B\right)}^{d}$ [41]. In this study, $v_{t}=\nabla_{S}^{D}\nabla^{d}r_{t}$, which was obtained by differentiating the series once regularly (d=1) and once seasonally (D=1). Thus, $v_{t}=\nabla_{24}^{1}\nabla^{1}r_{t}=\nabla_{24}^{1}w_{t}=w_{t}-w_{t-24}$.
(2)$$\begin{array}{cc} \Phi_{P}\left(B^{S}\right)\phi_{p}\left(B\right)v_{t} & =\Theta_{Q}\left(B^{S}\right)\theta_{q}\left(B\right)\varepsilon_{t},\mathrm{where},\\ \phi_{p}\left(B\right) & =1-\phi_{1}B-\cdots -\phi_{p}B^{p}\\ \theta_{q}\left(B\right) & =1+\theta_{1}B+\cdots +\theta_{q}B^{q}\\ \Phi_{P}\left(B^{S}\right) & =1-\Phi_{1}B^{S}-\Phi_{P}B^{PS}\\ \Theta_{Q}\left(B^{S}\right) & =1+\Theta_{1}B^{S}+\cdots +\Theta_{Q}B^{QS} \end{array}$$

For each stage of the time series, a common seasonal ARIMA model was fitted for the 27 observed time series (see Table 1). Firstly, each observed stage of the time series T was checked to determine if it could be regarded as stationary, which implies that the mean and variance are constant over time *t* and the covariance between one observation and another (in lagged *l* steps) from the same time series does not depend on *t* [44]. The ACF and PACF correlograms were used to examine this condition. Furthermore, the Augmented Dickey–Fuller (ADF) test [56] was applied for checking the null hypothesis of non stationarity, as well as the Lagrange multiplier (LM) test [57] for examining the null hypothesis about the absence of ARCH effect. In addition, the autocorrelation Ljung–Box Q (LBQ) test [58] was applied for inspecting the null hypothesis of independence in a given time series. This LBQ test was carried out on the different lags from nt to nt+48, where nt is the sum of the number of AR, MA, SAR and SMA terms of the seasonal ARIMA models. It was applied to the time series of the model residuals and the squared residuals.

The condition of stationarity is necessary when fitting an ARMA model. For this purpose, logarithmic transformation, one regular differentiation (d=1) and one seasonal differentiation (D=1) were applied to all time series of T, in order to stabilize the variance and remove both the trend in mean and seasonal trend [59]. Seasonal differentiation was applied to the observed time series W, and the results were denoted as V ($v_{t}=w_{t}-w_{t-24}$), being $w_{t}=r_{t}-r_{t-1}$.

In order to determine the appropriate values of (p,d,q) and (P,D,Q), the corrected Akaike’s Information Criterion (AICc) [49] was used, which checks how well a model fits the time series using the restriction d=1 and D=1. The most successful model for each stage of the different observed time series T was chosen according to the lowest AICc value. Next, the maximum likelihood estimation method was used to estimate the parameters of the seasonal ARIMA models [44]. Different tests were used to determine whether model assumptions were fulfilled. After computing the model residuals, ADF and LBQ [58] tests were applied to the residuals and their squared values, for 48 lags, in order to evaluate the condition of WN process. The ACF and PACF correlograms were also used. The next step was to evaluate the absence of Arch effects in the residuals. For this purpose, the LM test was applied to the residuals and their squared values [60,61]. Although the normality of errors is not an assumption for fitting ARIMA models, the distribution of residuals derived from the fitted models were compared with the normal distribution by means of the Q-Q normal scores plots, as well as the SW and KS normality tests.

Given that the errors of all models cannot be regarded as WN in this case, it is possible that the model residuals contain useful information about the performance of the different time series. In order to extract further information from the residuals, some features were calculated using ACF, PACF and statistics of normality tests, among others. They were used as additional classification variables.

The steps involved in Method 3 are illustrated in Figure 6. Firstly, the different time series were split according to the climatic stages observed (Data 2) and the structural breaks (SB) identified (Data 3). Secondly, the logarithm transformation was applied to Data 3 (the result is denoted as Data 4). The third step consisted of applying the seasonal ARIMA model to Data 4 in order to obtain the estimates of model coefficients: $\phi_{p}\left(B\right)$, $\theta_{q}\left(B\right)$, $\Phi_{P}\left(B^{S}\right)$ and $\Theta_{Q}\left(B^{S}\right)$. These parameters were denoted as Type 6 variables. Next, different features (Type 7 variables) were computed from the residuals: variance ($\sigma^{2}$), maximum of periodogram ($M_{I(w)}$) and its frequency *w*. For the set of 72 lags, additional features were computed from sample ACF values: mean ($\mu_{{\hat{\rho }}_{l}}$), median ($Md_{{\hat{\rho }}_{l}}$), variance ($\sigma_{{\hat{\rho }}_{l}}^{2}$) and range ($R_{{\hat{\rho }}_{l}}$), with l=1,⋯,72. Finally, the first four values of sample PACF ($\alpha_{1}$, $\alpha_{2}$, $\alpha_{3}$ and $\alpha_{4}$) were computed, as well as the statistics of the KS normality (Dn) and SW (Wn) tests.

In order to choose a seasonal ARIMA model and the estimations of the model parameters for each sensor, the arima (stats) and auto.arima (forecast) functions [33,34] were used. The ADF test was computed using the adf.test (aTSA) [32]. The LBQ test was applied by means of the Box.test function (stats). The LM test was carried out using the arch.test function (aTSA). The SW and KS normality tests were applied using the shapiro.test (stats) and ks.test functions (dgof), respectively.

#### 3.3.5. Calculation of Classification Variables—Method M4

The Wold decomposition establishes that any covariance stationary process can be written as the sum of a non-deterministic and deterministic process. This decomposition, which is unique, is a linear combination of lags of a WN process and a second process whose future values can be predicted exactly by some linear functions of past observations. If the time series $\left\{v_{t};t\in Z\right\}$ is purely non-deterministic, then it can be written as a linear combination of lagged values of a WN process (MA(*∞*) representation), that is, $v_{t}=\sum_{j=0}^{\infty }\psi_{j}\varepsilon_{t-j}$, where $\psi_{0}=1$, $\sum_{j=1}^{\infty }\psi_{j}^{2}<\infty$ and $\varepsilon_{t}$ is a WN [41]. Although the Wold decomposition depends on an infinite number of parameters, the values of coefficients of the decomposition decay rapidly to zero.

For Method 3, a unique model was fitted for the 27 sensors in the same stage of the time series because it is necessary to have the same number of classification variables per sensor, in order to apply later the sPLS-DA method. It is not possible to work with ’the best model’ per sensor in each stage. In order to obtain the same number of variables using the ’best model’ per sensor, the Wold decomposition was applied to each sensor. Hence, Method 4 consists of obtaining the Wold decomposition for the ARMA models. Firstly, different seasonal ARIMA models were fitted iteratively to time series $r_{t}$ per sensor and stage of the time series, and the most successful model was determined.

As an illustration, consider that a time series $r_{t}$ follows a seasonal ARIMA$\left(2,1,1\right){\left(0,1,0\right)}_{24}$ process. Now, consider that $w_{t}=r_{t}-r_{t-1}$ and $v_{t}=w_{t}-w_{t-24}$, with $t=1,\cdots ,t_{max}$. Then, the time series $v_{t}$ follows an ARMA(2,1) process, which can be decomposed according to the Wold approach by obtaining the polynomials $\phi_{p}\left(B\right)$ and $\theta_{q}\left(B\right)$ that determine the best ARMA (p,q) model. In summary, for each seasonal ARIMA $\left(p,d,q\right){\left(P,D,Q\right)}_{S}$, it was possible to find the best ARMA (p,q) model and its Wold decomposition.

The analysis of residuals of the different models fitted suggests that the condition of not autocorrelation is not fulfilled in all cases. Nonetheless, the Wold decomposition of each model was fitted independently, in order to have the ’best seasonal ARIMA model’ per sensor and the same number of parameters per sensor. For each model, the first five coefficients of the Wold decomposition were calculated and used as classification variables. In all cases, the most successful seasonal ARIMA model per sensor used D=1 and d=1.

The steps involved in Method 4 are illustrated in Figure 7. Firstly, the different time series were split according to the climatic stages observed (Data 2) and the structural breaks (SB) identified (Data 3). Secondly, the logarithm transformation was applied to Data 3 (the result is denoted as Data 4). The third step consisted of applying the seasonal ARIMA model to Data 4 in order to obtain the estimates of parameters and their residuals. Next, the same formulas of Type 7 variables used in M3 were applied to the residuals. Finally, the Wold decomposition was determined using the estimates of parameters of seasonal ARIMA models. The first five coefficients of the MA weights (i.e., $\psi_{1},\cdots ,\psi_{5}$) were denoted as Type 8 variables.

Apart from the same functions used for M3, this method employed ARMAtoMA (stats). One function was created for reducing a polynomial with AR and SAR components to a polynomial with just AR component. Likewise, another function converted a polynomial with MA and SMA components to one with MA component.

#### 3.3.6. Determination of Number of Classes and Class per Sensor Using PCA and K-Means Algorithm

All classification variables calculated as described above for each sensor were arranged as a matrix called ’total classification dataset’ (TCD) with 27 rows (sensors) by 671 columns corresponding to the classification variables from the four methods. The total number of variables was 88, 296, 143 and 139 from M1, M2, M3 and M4, respectively. The multivariate analysis of the TCD matrix would allow the identification of different microclimates in the archaeological museum. It was checked that the statistical distribution of some classification variables was strongly skewed, which recommends to apply a data pretreatment prior to the multivariate analysis. For those variables with a strongly skewed distribution, different standard (simple) Box–Cox transformations [62] were applied with the goal of finding a simple transformation leading to a normal distribution. In particular, the Box–Cox transformations were used on those classification variables with a Fisher coefficient of kurtosis [63] or with a Fisher–Pearson standardized moment coefficient of skewness [63] outside the interval from −2.0 to 2.0. Before applying the Box–Cox transformations, an absolute value function was used for variables with a negative skewness that fulfilled one of the aforementioned conditions. The skewness statistic was computed for each variable in order to check the asymmetry of the probability distribution. The kurtosis parameter indicates which variables were heavy-tailed or light-tailed, relative to a normal distribution. Moreover, the estimates of kurtosis were useful measures for identifying outliers in the classification variables. The functions kurtosis and skewness (PerformanceAnalytics) [38] were used to compute the coefficients of kurtosis and skewness, while boxcoxfit (geoR) [64] was employed to apply different Box–Cox transformations. The function prcomp (stats) was used to carry out PCA.

Those values of a given classification variable that clearly departed from a straight line on the normal probability plot were removed and regarded as missing data. These were estimated using the NIPALS algorithm [65] implemented in the mixOmics package [31], which is able to cope with this drawback and returns accurate results [66]. After the data normalization, all variables were mean-centered and scaled to unit variance, which is the common pretreatment in PCA. Next, PCA was carried out to reduce the dimensionality of the TCD matrix. Each observation (sensor) was projected onto the first few principal components to obtain lower-dimensional data, while preserving as much of the data variation as possible.

Given that the two first components maximize the variance of the projected observations (TCD), only two components were employed to run the k-means clustering. This method is a centroid-based algorithm that computes the distance between each sensor and the different centroids, one per cluster or class. The algorithm determines *K* clusters so that the intra-cluster variation is as small as possible. However, prior to applying this method, the number of clusters *K* has to be determined, which depends on the type of clustering and previous knowledge about the time series of T. For this purpose, different criteria can be used [39,67]. Such methods do not always agree exactly in their estimation of the optimal number of clusters, but they tend to narrow the range of possible values. The NbClust function of the NbClust package [39] incorporates 30 different indices for determining the number of clusters [67]. This function claims to use the best clustering scheme from the different results obtained, by varying all combinations of the number of clusters, distance measures and clustering methods. It allows the user to identify the value *K* in which more indices coincide, providing assurance that a good choice is being made.

In this clustering method, the measure used to determine the internal variance of each cluster was the sum of the squared Euclidean distances between each sensor and each centroid. The distances were used to assign each sensor to a cluster. For this purpose, the k-means algorithm of Hartigan and Wong [68] was applied by means of the function kmeans (stats). It performs better than the algorithms proposed by MacQueen [69], Lloyd [70] and Forgy [71]. However, when the algorithm of Hartigan and Wong is carried out, it is often recommended to try several random starts. In the present study, 100 random starts were employed. This algorithm guarantees that, at each step, the total intra-variance of the clusters is reduced until reaching a local optimum. Results from the k-means algorithm depend on the initial random assignment. For this reason, the algorithm was run 100 times, each with a different initial assignment. The final result was the one leading to a classification with the lowest total variance value. By comparing the classification obtained with the position of sensors in the museum (Figure 8a), the three zones were denoted as North West (NW), South East (SE) and Skylight (Sk).

#### 3.3.7. Sensor Classification Using sPLS-DA

Partial Least Squares (PLS) regression [72] is a multivariate regression method which relates two data matrices (predictors and answer). PLS maximizes the covariance between latent components from these two datasets. A latent component is a linear combination of variables. The weight vectors used to calculate the linear combinations are called loading vectors.

Penalties such as Lasso and Ridge [73] have been applied to the weight vectors in PLS for variable selection in order to improve the interpretability when dealing with a large number of variables [74,75,76]. Chung and Keles [77] extended the sparse PLS [76] to classification problems (SPLSDA and SGPLS) and demonstrated that both SPLSDA and SGPLS improved classification accuracy compared to classical PLS [78,79,80]. Le Cao et al. [81] introduced a sparse version of the PLS algorithm for discrimination purposes (sPLS-Discriminant Analysis, sPLS-DA) which is an extension of the sPLS proposed by Lê Cao et al. [74,75]. They showed that sPLS-DA has very satisfying predictive performances and is able to select the most informative variables. Contrary to the two-stages approach (SPLSDA) proposed by Chung and Keles [77], sPLS-DA performs variable selection and classification in a single-step procedure. In order to classify the sensors and improve the interpretability of results, sPLS-DA was applied to the different classification datasets.

Since the original PLS algorithm proposed by Wold [72], many variants have arisen (e.g., PLS1, PLS2, PLS-A, PLS-SVD [82] and SIMPLS [83]), depending on how the regressor matrix (X) and response matrix (Z) are deflated. Alternatives exist whether X and Z are deflated separately or directly, using the cross product $M=X^{⊤}Z$ and the Singular Value Decomposition (SVD). A hybrid PLS with SVD is used in the version sPLS-DA [74]. For sPLS-DA, a regressor matrix is denoted as X ($X_{1}$, $X_{2}$, $X_{3}$ or $X_{4}$ in this case), with dimension n×p. The number of rows (sensors) is n=27 and the columns correspond to the classification variables *p* ($p_{1}$, $p_{2}$, $p_{3}$ or $p_{4}$). A response qualitative vector denoted as Y has length *n* and it indicates the class of each sensor, with values coded as 1 (for NW), 2 (SE) and 3 (Sk).

sPLS-DA was carried out using Lasso penalization of the loading vectors associated to X [84] using a hybrid PLS with SVD decomposition [85]. The penalty function is included in the objective function of PLS-DA, which corresponds to PLS carried out using a response matrix Z with values of either 0 or 1, created with the values of response vector Y. Thus, this vector was converted into a dummy matrix Z with dimension n×K, being n=27 the number of sensors and K=3 the number of sensor classes.

Regarding the optimization problem of sPLS-DA, in this case: $X\in R^{n\times p}$ is a matrix with *p* variables and *n* sensors, $Y\in R^{n\times 1}$ is a response vector (classes k=1,2,3) and $Z\in {\left\{0,1\right\}}^{n\times 3}$ is an indicator matrix, where $z_{ik}=I\left(Y_{i}=k\right)$, with k=1,2,3. The sPLS-DA method modeled Z and X as $X=\Xi C+E_{1}$ and $Z=\Xi D+E_{2}$, where C and D are matrices that contain the regression coefficients of X and Z on the *H* latent components associated to X, while $E_{1}$ and $E_{2}$ are random errors. Furthermore, each component of Ξ is a combination of selected variables, where $\Xi =[\xi_{1},\ldots ,\xi_{H}]$, and each vector $\xi_{h}$ was computed sequentially as $\xi_{h}=X_{h-1}u_{h}$, where $X_{h-1}$ is the orthogonal projection of X on subspace $span{\left\{\xi_{1},\ldots ,\xi_{h-1}\right\}}^{⊥}$ and $\left(u_{h},v_{h}\right)$ is the solution the optimization problem according to Equation (Equation 3), subject to $∥u_{h}{∥}_{2}=1$.
(3)$$\left(u_{h},v_{h}\right)=\underset{u_{h},v_{h}}{arg min}\{∥M_{h}-u_{h}v_{h}^{⊤}{∥}_{F}^{2}+P_{\lambda }\left(u_{h}\right)\}$$

The optimization problem minimizes the Frobenius norm between the current cross product matrix ($M_{h}$) and the loading vectors ($u_{h}$ and $v_{h}$), where $M_{h}=X_{h}^{⊤}Z_{h}$ and $Z_{h-1}$ is the orthogonal projection of Z on subspace $span{\left\{\xi_{1},\ldots ,\xi_{h-1}\right\}}^{⊥}$. Furthermore, $∥M_{h}-u_{h}v_{h}^{⊤}{∥}_{F}^{2}=\sum_{i=1}^{n}\sum_{j=1}^{p}{\left(m_{ij}-u_{i}v_{j}\right)}^{2}$, and $P_{\lambda }\left(u_{h}\right)$, defined as $\lambda ∥u_{h}{∥}_{1}$, is the Lasso penalty function [75,81]. This optimization problem is solved iteratively based on the PLS algorithm [86]. The SVD decomposition of matrix $M_{h}$ is subsequently deflated for each iteration *h*. This matrix is computed as $M_{h}=u\Delta v^{⊤}$, where u and v are orthonormal matrices and Δ is a diagonal matrix whose diagonal elements are called the singular values. During the deflation step of PLS, $M_{h}\neq X_{h}^{⊤}Z_{h}$, because $X_{h}$ and $Z_{h}$ are computed separately, and the new matrix is called ${\tilde{M}}_{h}$. At each step, a new matrix ${\tilde{M}}_{h}=X_{h}^{⊤}Z_{h}$ is computed and decomposed by SVD [74]. Furthermore, the soft-thresholding function $g\left(u\right)={\left(|u|-\lambda \right)}_{+}sign\left(u\right)$, with ${\left(x\right)}_{+}=max\left(0,x\right)$, was used in penalizing loading vectors u to perform variable selection in regressor matrix, thus $u_{new}=g_{\lambda }\left({\tilde{M}}_{h-1}v_{old}\right)$ [74].

Rohart et al. [30] implemented an algorithm to solve the optimization problem in Equation (Equation 3), where the parameter λ needs to be tuned and the algorithm chooses λ among a finite set of values. It is possible to find a value λ for any null elements in a loading vector. These authors implemented the algorithm using the number of non-zero elements of the loading vector as input, which corresponds to the number of selected variables for each component. They implemented sPLS-DA in the R package mixOmics, which provides functions such as perf, tune.splsda and splsda, in order to determine the number of components and elements different to zero in the loading vector before running the final model.

In order to compare the performance of models constructed with a different number of components, 1000 training and test datasets were simulated and the sPLS-DA method (for a maximum number of 10 components) was tuned by three-fold cross-validation (CV) for X. The perf function outputs the optimal number of components that achieve the best performance based on both types of classification error rate (CER): Balanced Error Rate (BER) and the Overall classification error rate (Overall). BER is the average proportion of wrongly classified sensors in each class, weighted by the number of sensors. In most cases, the results from sPLS-DA were better (or very similar) using Overall than when using BER. However, BER was preferred to Overall because it is less biased towards majority classes during the performance assessment of sPLS-DA. In this step, three different prediction distances were used, maximum, centroid and Mahalanobis [30], in order to determine the predicted class per sensor for each of the test datasets. For each prediction distance, both Overall and BER were computed.

The maximum number of components found in the present work was three (K=3) when BER was used instead of Overall. Furthermore, when sPLS-DA used classification variables from M2 and M3, two components led to the lowest BER values. Among the three prediction distances calculated (i.e., maximum, centroid and Mahalanobis), it was found that centroid performed better for the classification. Thus, this distance was used to determine the number of selected variables and to run the final model. Details of distances can be found in the Supplementary Materials of [30].

In order to compare the performance of diverse models with different penalties, 1000 training and test datasets were simulated and the sPLS-DA method was carried out by three-fold CV for X. The performance was measured via BER, and it was assessed for each value of a grid (keepX) from Component 1 to *H*, one component at a time. The different grids of values of the number of variables were carefully chosen to achieve a trade-off between resolution and computational time. Firstly, a two coarse tuning grids were assessed before setting a finer grid. The algorithm used the same grids of keepX argument in tune.splsda function to tune each component.

Once the optimal parameters were chosen (i.e., number of components and number of variables to select), the final sPLS-DA model was run on the whole dataset X. The performance of this model in terms of BER was estimated using repeated CV.

In summary, three-fold CV was carried out for a maximum of ten components, using the three distances and both types of CER. The optimal number of components was obtained by using the BER and centroid distance. Next, the optimal number of variables was identified by carrying out the second three-fold CV. It was run using BER, centroid distance and values of three grids with different number of variables. Next, when both optimal numbers were obtained, the final model was computed.

Regarding the most relevant variables for explaining the classification of sensors, there are many different criteria [86,87,88]. The first measure selected is the relative importance of each variable for each component and another is the accumulated importance of each variable from components. Both measures were employed in this research.

Lê Cao et al. [81] applied sPLS-DA and selected only those variables with a non-zero weight. The sparse loading vectors are orthogonal to each other, which leads to uniquely selected variables across all dimensions. Hence, one variable might be influential in one component, but not in the other. Considering the previous argument and that the maximum number of components was three (*h* = 1, 2, 3), Variable Importance in Projection (VIP) [86] was used to select the most important ones. It is defined using loading vectors and the correlations of all response variables, for each component. $VIP_{hj}$ denotes the relative importance of variable $X_{j}$ for component *h* in the prediction model. Variables with $VIP_{hj}>1$ are the most relevant for explaining the classification of sensors. $VIP_{hj}$ was calculated using the vip function (mixOmics). Although the assumption of the sparse loading vectors being orthogonal was considered, in practice, some selected variables were common in two components. Then, a second measure of VIP [88] was employed: $VIP_{j}$ denotes the overall importance of variable $X_{j}$ on all responses (one per class) cumulatively over all components. It is defined using the loading vectors and the sum of squares per component. Variables with $VIP_{j}>1$ are the most relevant for explaining the classification of sensors. The selected variables were ranked according to both types of VIPs, which are discussed below for each stage of the time series.

#### 3.3.8. Sensor Classification Using Random Forest Algorithm

The RF algorithm [89] handles big datasets with high dimensionality. It consists of a large number of individual decision trees that were trained with a different sample of classification variables (X) generated by bootstrapping. The overall prediction from the algorithm was determined according to the predictions from individual decision trees. The class which receives most of the votes was selected as the prediction from each sensor. In addition, it can be used for identifying the most important variables.

An advantage of using the bootstrap resampling is that random forests have an Out-Of-Bag (OOB) sample that provides a reasonable approximation of the test error, which allows a built-in validation set that does not require an external data subset for validation. The following steps were carried out to obtain the prediction (output) from the algorithm, using the different classification data. Firstly, from a classification dataset, *B* random samples with replacements (bootstrap samples) were chosen, as well as one sample of 17 sensors. Next, from each bootstrap sample, a decision tree was grown. At each node, *m* variables out of total *p* were randomly selected without replacement. Each tree used an optimal number of variables that was determined by comparing the OOB classification error of a model, based on the number of predictors evaluated. Each node was divided using the variable that provided the best split according to the value of a variable importance measure (Gini index). Each tree grew to its optimal number of nodes. The optimal value of this hyper-parameter was obtained by comparing the OOB classification error of a model, based on the minimum size of the terminal nodes. From each bootstrap sample, a decision tree came up with a set of rules for classifying the sensors. Finally, the predicted class for each sensor was determined using those trees which excluded the sensor from its bootstrap sample. Each sensor was assigned to the class that received the majority of votes.

If the number of trees is high enough, the OOB classification error is roughly equivalent to the leave-one-out cross-validation error. Furthermore, RF does not produce overfitting problems when increasing the number of trees created in the process. According to previous arguments, 1500 trees were created for all cases to run the algorithm and the OOB classification error was used as an estimate of the test error. Prior to running the final RF algorithm, the optimal values of the number of predictors and the terminal nodes were determined (i.e., those corresponding to a stable OOB error).

In order to select the most important variables, the methodology proposed by Han et al. [90] was used, which is based on two indices: Mean Decrease Accuracy (MDA) and Mean Decrease in Gini (MDG). The combined information is denoted as MDAMDG. Some reasons for using the methodology were: (1) The OOB error usually gives fair estimations compared to the usual alternative test set error, even if it is considered to be a little bit optimistic. (2) The use of both indices is more robust than considering any individual one [90]. MDA corresponds to the average of the differences between OOB error before permuting the values of the variable and OOB error after permuting the values of the variable for all trees. Because a random forest is an ensemble of individual decision trees, the expected error rate called Gini impurity [91] is used to calculate MDG. For classification, the node impurity is measured by the Gini index, and the MDG index is based on this. The former is the sum of a variable’s total decrease in node impurity, weighted by the proportion of samples reaching that node in each individual decision tree in the random forest [92]. The higher is the MDG, the greater is the contribution of a given variable in the classification of sensors. The functions randomForest and importance [93] were used to carried out the RF algorithm.

## 4. Results and Discussion

The methodology employed consists of using sPLS-DA and RF in order to classify time series of T and to determine the optimal variables for discriminating the series. Both methods had input features calculated from different functions (e.g., sample ACF, sample PACF, spectral density and MR), as well as estimated parameters from the seasonal ARIMA model, Wold decomposition and the last level of smoothing components from the additive SH-W method. Additionally, other features were computed such as the mean, variance and maximum values of functions applied to residuals of the seasonal ARIMA or SH-W models (e.g., sample ACF, sample PACF, spectral density and statistics of the SW and KS normality test, among others). For sPLS-DA, centroid distance and BER were considered when running the final model. Two indicators of VIPs were used to rank the selected variables. The first one measures the relative importance of each relevant variable, per component in the prediction model, while the second indicator evaluates the overall importance of each variable over all components. Additionally, based on the results from sPLS-DA, a methodology was proposed to reduce the number of sensors required for a long-term microclimate monitoring. Regarding RF, the OOB classification error was used as an estimate of the test error and the parameters MDG and MDA were computed as indicators of the variable importance. Both sPLS-DA and RF methods were applied to select and compare the optimal variables that explain the classification of sensors according to values of T.

Before carrying out sPLS-DA and RF, the k-means algorithm with PCA was employed, in order to characterize the different zones in the museum, according to the indoor temperature. Once the three zones (NW, SE and Sk) were established, these classes were used as input for sPLS-DA and RF.

### 4.1. Identification of Structural Breaks in the Time Series

The supF and CUSUM tests were applied to study the stages. In Wr1, the former test revealed a structural break after the 682nd observation (20 November at 7:00 a.m., *p*-value = 0.01). In Cd, this test suggests another break after the 1981st observation (13 January at 10:00 a.m., *p*-value = 0.02). Finally, a structural break was also found in the Ht stage at the 6133rd observation (4 July at 10:00 a.m., *p*-value = 0.03). Accordingly, the stages Wr1, Cd and Ht were split in two parts (i.e., before and after the structural break). Thus, the following eight stages of T were considered: Wr1A (warm 1 before the structural break, n = 682), Wr1B (warm 1 after the break, n = 808), CdA (cold period before the break, n = 491), CdB (cold stage after the break, n = 1212), Tr (transition, n = 2303), HtA (hot stage before the break, n = 637), HtB (hot stage after the structural break, n = 1690) and Wr2 (warm 2, n = 903).

The CUSUM chart identified a significant shift at the same instances for the stages Wr1, Cd and Ht. The main reason for these structural breaks could have been sudden changes of T outside the museum or possibly modifications in the air conditioning and heating systems. January is usually the coldest month of the year in Valencia, while July and August are the hottest ones. It is reasonable to assume that the configuration of the air conditioning system was modified in these months, in order to maintain an appropriate microclimate inside the archaeological site.

According to the structural breaks, all time series were split into eight stages, and the four methods explained in the next sections were applied to each stage. This step is necessary to avoid possible problems with the properties of the estimated parameters of the models applied [94]. The four methods (M1–M4) were carried out separately for each stage of the 27 time series of T or W: Wr1A, Wr1B, CdA, CdB, Tr, HtA, HtB and Wr2. As an exception, in Method 2, apart from modeling each stage separately, the complete time series was also considered.

### 4.2. Calculation of Classification Variables—M1–M4


For Method 3, for the most successful seasonal ARIMA models, both tests (KS and SW) rejected the normality hypothesis of the errors in 100% of cases. Furthermore, all Q-Q normal score plots displayed that residuals were not falling close to the line at both extremes. ADF test suggests that the errors are stationary in all cases. According to the LBQ test, the errors are independent (lags from nt to nt+48) as maximum in 77.00% of the 27 sensors (in stage Wr1A) and as minimum 0.0% (HtB). The LM test suggests the absence of Arch effects as maximum 92.52% (Wr1A) and as minimum 3.70% (Wr1B) (see Table 1).

For Method 4, for the most successful seasonal ARIMA models (see Table 2), both tests of normality rejected the hypothesis of normal distribution for the errors in 100% of models. The ADF test suggests that errors are stationary in all cases. The analysis of residuals of the different fitted models indicates that the condition of not autocorrelation (WN) is not fulfilled in all cases. In particular, according to the LBQ test, errors are autocorrelated (up to lag 24) at least in 25.93% of the models (for stage HtB) and the maximum was 92.59% (Wr1A). The LM test suggests absence of Arch effects, with a maximum of 96.30% (Wr1A) and a minimum of 3.70% (Tr) (see Table 1). In order to extract further information, the same features computed in M3 for the residuals (Type 7 variables) were also estimated in M4 and used as classification variables.

The results of Methods 1–4 were arranged as matrices, denoted as $X_{1}$ (27 sensors × 88 variables), $X_{2}$ (27 sensors × 296 variables), $X_{3}$ (27 sensors × 143 variables) and $X_{4}$ (27 sensors × 139 variables), respectively. The regressor matrices $X_{1}$ to $X_{4}$ contained the following percentages of missing values: 5.28%, 5.66%, 3.55% and 8.61%, respectively.

### 4.3. Determination of Number of Classes and Class per Sensor Using PCA and K-Means Algorithm

The best option appeared to be K=6 (coincidence of 8 out of the 27 indices), followed by K=2 and K=3 (coincidences of 7 and 5, respectively), and then K=4 and K=5 (Figure 8a). According to previous research, there is a pronounced temperature gradient at the museum, particularly in summer, caused by the greenhouse effect of the skylight. Then, each cluster or class was expected to be related to the distance of each sensor from the skylight, among other factors, such as the influence of the weather conditions outdoors and the effect of the air conditioning system. Furthermore, due to the large size of the museum (2500 $m^{2}$) and the temperature gradient that exists at the entrance and below the skylight, it seems better to consider three zones instead of just two. Hence, K=3 was the number of clusters used for the k-means algorithm.

The k-means method classified sensors C0, C1, A4, A5 and F in the SE zone. However, by checking their position on the map of the museum (Figure 8b), these sensors could be regarded in the boundary between the NW and SE zones. Hence, it should be discussed whether such classification is appropriate, or if they should be regarded within the NW zone. In order to study this issue, two classifications were analyzed using sPLS-DA: (1) by considering A4, A5 and F in the NW zone; and (2) by locating C0 and C1 in the NW zone. For both cases, the rate of misclassified sensors was computed by means of sPLS-DA. The error rates for the classification from the k-means algorithm were: 0.25 (M1), 0.30 (M2), 0.29 (M3) and 0.42 (M4). For Case (1), the classification error rates were: 0.15 (M1), 0.19 (M2), 0.31 (M3) and 0.35 (M4). For Case (2), the error rates were: 0.21 (M1), 0.23 (M2), 0.30 (M3) and 0.41 (M4). It turns out that the lowest error rates were found for Case (1). Thus, sensors A4, A5 and F were considered as part of the NW zone for the next sections. Finally, 13 sensors were classified in the NW zone (A4, A5, A6, B5, B6, C5, C6, D1, D2, D3, D5, D6 and F), eight in the SE zone (A, B, B1, C, C0, C1, D and G) and six in the Sk zone (A2, A3, B2, B3, B4 and C3). The proposed classification of sensors is shown in Figure 1a, which depicts an association between T values and the three zones of the museum: NW, SE and Sk.

### 4.4. Sensor Classification Using sPLS-DA

Figure 9 displays the results from the first three-fold CV for the four methods (M1–M4) and when using variables from all methods (Ms). According to the results, centroid distance performed the best for the first two components, in the case of M1, M4 and Ms. For M1 and Ms, Overall or BER performed the best for both classification error rates. For the other methods (M2, M3 and M4), Overall was the best. The centroid distance and BER from these results were selected as input in the next step of the method (see Figure 10). The values of BER and centroid distance suggested that three components are enough to classify the sensors. Figure 10 shows the results from the second three-fold CV for the final grid (5, 10 and 15 variables). The results suggest that the number of variables for the first component of each method were the following: 15 (M1 and M2), 5 (M3 and M4) and 15 for Ms. The information displayed in Figure 9 and Figure 10 (centroid distance, BER and number of variables per component) was used to apply the final model.

The notation used in this study are the estimates of the following parameters: mean, range, median and variance of ACF values (acf.m, acf.r, acf.md and acf.v); PACF at lags 1–6 (pacf1–pacf6); mean of the time series (M); statistics of the KM test (kolg.t); statistics of the SW test (shap.t); maximum values of the periodogram (spec.mx); frequency of the maximum values of the periodogram (freq); variance of the residuals (res.v); SSE (sse); seasonal components (s1–s24); level (a); slope (b); coefficients of the Wold decomposition (psi1–psi5); the first AR term (ar1); the first MA term (ma1); the first SMA term (sma1); the first SAR term (sar1); and the 17th prediction of T, which was denoted as pred.17.

The most relevant variables per method are the estimates of the following parameters:☐Selected variables from M1: PACF at lags 1–4, MR and parameters of the sample ACF (mean, range, median and mean) (Table 3 (a)).☐Selected variables from M2: Level, slope, some seasonal components (5, 8, 10–12, 16–21 and 23), mean and median of sample ACF (residuals), SSE (residuals) and maximum of the periodogram (residuals) (Table 3 (b)).☐Selected variables from M3: Parameters of MA, SAR and SMA of seasonal ARIMA models, PACF at lags 1–5 (residuals), mean and median of sample ACF (residuals), statistic of KM normality test (residuals) and frequency for the maximum of the periodogram (residuals) (Table 4 (a)).☐Selected variables from M4: The first four Wold coefficients, mean and median of sample ACF (residuals), statistics of the SW and KM normality tests (residuals), PACF at first five lags (residuals), variance (residuals) and maximum of the periodogram (residuals) (see Table 4 (b)).☐Selected variables when using the total set of variables from the four methods (some values are highlighted in bold and blue in Table 3 and Table 4): Slope (M2), some seasonal components (2, 3, 5, 8, 11, 12, 14, 16–18 and 21–23) (M2); SSE (M2); Wold Coefficient 1 (M4); mean (M1–M4), median (M2), range (M1 and M2) and variance (M1 and M2) of sample ACF values; and maximum of the periodogram (M2–M4).

**Table 3 sensors-21-04377-t003:** Selected variables (V) per component (C) from sPLS-DA. Variables highlighted in bold and blue correspond to selected variables for Ms.

	C1		C2		C3	
	**Stage**	**V**	**Stage**	**V**	**Stage**	**V**
**(a)** For Method 1						
1	Tr	**pacf2**	Wr2	**acf.r**	CdA	pacf3
2	Tr	**acf.m**	Wr2	**acf.v**	Wr1A	pacf4
3	HtA	**pacf2**	HtA	**acf.m**	CdA	pacf4
4	HtA	**pacf1**	CdB	pacf1	Wr1A	acf.r
5	HtB	**pacf1**	CdB	M	Wr1A	acf.v
6	CdB	pacf2	Wr1B	M	Wr1A	rMh
7	Wr2	pacf1	HtB	M	Wr1B	rMh
8	Wr1B	pacf1	Wr1A	M	CdA	rMh
9	CdA	pacf1	CdA	pacf1	CdB	rMh
10	HtB	pacf2	Wr1B	acf.r	Tr	rMh
11	Wr1B	pacf3	Wr1B	acf.v	HtA	rMh
12	Tr	M	Wr2	pacf2	HtB	rMh
13	HtA	pacf4	CdA	pacf2	Wr2	rMh
14	Tr	pacf4	HtA	acf.r		
15	Wr2	acf.m	HtA	acf.v		
**(b)** For Method 2						
1	HtA	**sse**	HtA	**acf.m**		
2	Tr	**sse**	HtA	**acf.md**		
3	Tr	**s18**	HtA	**s12**		
4	Tr	**b**	HtA	**s11**		
5	Tr	**s16**	HtA	**acf.r**		
6	HtB	**sse**	CdA	**acf.v**		
7	Tr	**s17**	HtA	**acf.v**		
8	HtB	**s21**	HtA	**s23**		
9	Tr	**spec.mx**	CdA	**s14**		
10	Wr2	s20	CdA	**s5**		
11	HtB	acf.m	Tr	**a**		
12	Wr1B	s12	CdA	s10		
13	Wr2	acf.md	HtA	**s8**		
14	HtB	s19	HtA	s10		
15	HtB	s23	Wr1A	acf.m		

**Table 4 sensors-21-04377-t004:** Selected variables (V) per component (C) from sPLS-DA. Variables highlighted in bold and blue correspond to selected variables for Ms.

	C1		C2		C3	
	**Stage**	**V**	**Stage**	**V**	**Stage**	**V**
**(a)** For Method 3						
1	Tr	**acf.m**	CdA	acf.m		
2	HtA	res.v	CdA	pacf4		
3	Tr	ma1	CdA	pacf5		
4	HtB	ma1	Wr2	pacf4		
5	Tr	kolg.t	Tr	sar2		
6			CdB	pacf2		
7			HtB	pacf1		
8			Wr2	sar2		
9			CdA	sar1		
10			HtA	sma1		
11			HtA	pacf3		
12			HtA	freq		
13			HtA	pacf1		
14			HtB	acf.md		
15			Wr1A	pacf5		
**(b)** For Method 4						
1	HtA	psi1	HtA	pacf5	Wr1B	**psi1**
2	HtA	spec.mx	HtB	pacf1	CdB	**acf.m**
3	HtA	res.v	Wr2	acf.v	CdB	**spec.mx**
4	Tr	psi1	HtA	shap.t	Wr2	acf.m
5	Wr1A	acf.m	HtA	acf.m	Wr2	shap.t
6			Wr2	acf.r		
7			Tr	pacf1		
8			Tr	acf.md		
9			Wr1B	psi4		
10			CdA	pacf5		
11			Tr	psi3		
12			CdA	acf.md		
13			Wr2	psi2		
14			Tr	kolg.t		
15			Wr2	kolg.t		

Based on the information in Table 3 and Table 4, the two most influential stages of the time series are highlighted in bold in Table 5 per component and method, or just one stage when it had a value greater than 50%. It can be deduced from the results in Table 5 that the two most important stages for classifying sensors were HtA and Tr, which is intuitively appealing because important temperature fluctuations occur in summer due to the greenhouse effect caused by the skylight.

Figure 11 displays the estimations of the average BER over all components from the sPLS-DA. According to the results, M1 obtained the best performance, followed by M2.

For Component 1 (C1) versus Component 2 (C2) from sPLS-DA: C1 displays a clear discrimination between sensors in the Sk class against NW and SE for both M1 and M2 (see Figure 12a,d). This fact is more clear for M2. For M2, C2 clearly separates Sk vs. SE, as well as Sk vs. NE (see Figure 12a). For M3, C1 discriminates Sk vs. NW satisfactorily, and C2 shows the same performance for this method (see Figure 12e). For M4, C1 properly separates SE vs. NW, while C2 discriminates between SE and Sk (see Figure 12d). When using all variables (Ms), C1 shows an adequate discrimination of Sk vs. NW and C2 between Sk vs. SE (see Figure 12i). For C1 vs. C3 from sPLS-DA: For M1, C1 clearly separates Sk against NW (see Figure 12b). For M4, C1 discriminates NW against the other classes, but Sk and SE appear overlaid (see Figure 12g). With C3, the classes could not be discriminated. For C2 vs. C3 from sPLS-DA: C2 properly separates SE against the other clusters in M1 and Ms, but the other classes appear overlaid (see Figure 12c,k). For M4, C2 shows a good discrimination of Sk against the other classes (see Figure 12h), but C3 did not yield any separation.

Figure 13 displays the results from sPLS-DA for M1. The scatter plot in Figure 13a shows the projection of sensors over Components 1 and 2. According to the three confidence ellipses (at a confidence level of 95%) and their centroids, each class can be characterized using the centroids. Furthermore, the bar plots in Figure 13b,c represent the loading weights of the selected variables for C1 and C2, respectively. A different color was used to indicate the class (SE, NW and Sk) in which the variable yields the maximum level of expression on average, which was computed to have an approximate assessment of the influence of variables in each class, per component. The selected variables that characterize each class were the following: for the NW class, the mean of values ACF and PACF at lags 1–3; for the SE zone, PACF at lags 1 and 2 and the mean of the time series; and for the Sk category, the mean, range and variance of ACF values and the mean of the time series.

### 4.5. Sensor Classification Using Random Forest Algorithm

Figure 11 displays the values of OOB classification error. The best results were achieved from Ms and the second best result from M1. Table 6 shows the selected variables in descending order, according to MDAMDG. The most relevant variables per method are the estimates of the following parameters:
☐Selected variables from M1: Mean, range and variance of the sample ACF values, the first four values of sample PACF, the maximum value of the periodogram and the mean.☐Selected variables from M2: Level, slope and 18 seasonal components (2, 3, 7–12, 14 and 16–24) from sH-W method, SSE, maximum of the periodogram values, statistic of the KS normality test and 17th prediction of T.☐Selected variables from M3: First term of the AR, MA and SMA components of the seasonal ARIMA models, sample PACF at lags 1, 2 and 5, mean and median of the sample ACF values, residual variance, maximum of the periodogram values and statistic of the KS normality test.☐Selected variables from M4: Coefficients 1, 3, 4 and 5 from the Wold decomposition; the first five values of the sample PACF; the mean, median, range and variance of the sample ACF values; residual variance; maximum of the periodogram values of the residuals; and statistics of both the SH and KS normality tests.☐Selected variables when using all classification variables from the four methods (Ms): mean (M1–M3), median (M2 and M3), range (M2) and variance (M2) of the sample ACF values; sample PACF at lags 1 (M1 and M3), 2 (M1) and 3 (M1); first coefficient of the Wold decomposition (M3); and statistic of the KS normality test (M2) (the results from Ms comprise 20% of variables from M1, 60% from M2, 9% from M3 and 11% from M4).

**Table 6 sensors-21-04377-t006:** Selected variables (V) from RF per method and when using the variables from the four methods.

	M1		M2		M3		M4		Ms		
	**Stage**	**V**	**Stage**	**V**	**Stage**	**V**	**Stage**	**V**	**M**	**Stage**	**V**
1	CdA	pacf1	HtA	sse	Tr	acf.m	HtA	psi1	M1	CdA	pacf1
2	HtA	pacf4	HtA	s24	HtA	sma1	CdB	psi1	M2	HtA	sse
3	HtA	pacf1	HtA	s23	Tr	acf.md	Tr	res.v	M3	Tr	acf.m
4	Tr	pacf2	HtA	s8	CdB	pacf2	HtA	pacf5	M2	HtA	s24
5	CdB	pacf2	HtA	s11	HtA	res.v	HtA	res.v	M4	HtA	psi1
6	HtA	acf.m	HtA	s12	Tr	kolg.t	Tr	acf.m	M1	HtA	pacf4
7	Tr	acf.m	HtB	acf.m	HtA	spec.mx	Tr	spec.mx	M2	HtA	s23
8	CdA	acf.m	CdA	acf.v	CdB	ma1	Tr	psi1	M3	HtA	sma1
9	Tr	pacf1	Tr	sse	Tr	ma1	Tr	kolg.t	M2	HtA	s8
10	HtB	pacf2	HtA	acf.r	HtA	pacf1	CdA	pacf1	M2	HtA	s11
11	HtB	M	HtA	acf.md	CdA	pacf5	HtA	spec.mx	M2	HtA	acf.r
12	Wr2	acf.m	Wr2	s23	HtA	ar1	Wr1A	acf.m	M2	HtA	s12
13	Wr2	pacf1	Wr1B	a	Wr1A	sar2	CdA	acf.md	M2	HtA	acf.m
14	CdB	pacf1	Tr	b	Wr1A	acf.md	CdA	pacf4	M2	CdA	acf.v
15	Tr	M	HtA	s22	Tr	spec.mx	CdA	acf.v	M2	Wr1B	a
16	Wr2	acf.v	Tr	s16	CdA	acf.m	HtB	res.v	M2	HtA	s22
17	Wr2	pacf2	Tr	a	Tr	res.v	CdA	kolg.t	M2	HtB	acf.m
18	HtA	pacf2	Tr	s17	HtB	ma1	Wr1A	psi1	M1	HtA	pacf1
19	Wr2	acf.r	CdA	s10	HtB	sar2	Wr1B	acf.m	M2	HtA	acf.md
20	CdA	M	CdA	a	HtB	pacf1	Tr	pacf1	M3	Tr	acf.md
21	Wr1B	M	HtA	acf.v	CdB	kolg.t	Tr	pacf5	M2	HtA	s9
22	HtB	pacf1	Wr2	s18	CdA	sar1	HtB	psi1	M2	Wr2	s23
23	Wr1B	pacf3	HtB	kolg.t	HtA	pacf3	Tr	acf.md	M2	Tr	s17
24	HtA	M	Wr2	s19	HtA	pacf5	Wr2	pacf4	M2	CdA	a
25	CdB	M	HtB	s21	Tr	freq	CdA	acf.m	M1	Tr	pacf2
26	Wr1A	pacf4	HtA	s10	Wr1B	acf.md	HtA	shap.t	M2	Tr	sse
27	HtB	pacf4	HtB	s9	Wr1A	ma1	HtA	acf.m	M2	HtA	acf.v
28	Wr1B	pacf2	Tr	spec.mx			Wr1B	psi4	M4	CdB	psi1
29	HtB	spec.mx	Tr	s18			Wr2	res.v	M2	HtA	s10
30	Wr1B	acf.m	CdA	s5			CdA	pacf5	M2	Tr	spec.mx
31	HtA	pacf3	Tr	s20			Wr2	acf.r	M3	CdB	pacf2
32	Wr1B	pacf1	CdA	s14			CdA	shap.t	M2	Tr	s16
33	CdA	pacf3	CdA	s1			Wr2	acf.v	M2	Tr	b
34	CdB	pacf4	HtA	acf.m			HtB	pacf1	M2	Wr2	s20
35	Wr1A	pacf2	HtA	s9			Wr2	spec.mx	M3	CdB	ma1
36	Wr1A	M	Wr2	s17			Wr1B	psi3	M2	CdA	s1
37	CdA	pacf2	All	pred.18			Wr2	pacf2	M2	CdA	s10
38	Tr	acf.v	CdA	s22			Wr1B	pacf5			
39	Wr2	pacf4	HtA	spec.mx			CdB	shap.t			
40	Tr	pacf4	Wr2	s3			HtA	psi5			
41	HtA	spec.mx	CdA	s23			Tr	shap.t			
42	CdB	acf.m	Wr2	s2			Wr1A	pacf5			
43	Tr	acf.r					Wr2	kolg.t			
44	HtB	acf.m					CdA	pacf3			

According to Table 7, the most important stages for classifying the time series were HtA and Tr. The same result was found from the sPLS-DA method. This result makes sense because there is a pronounced temperature gradient inside the museum, due to the skylight.

For the purpose of time series discriminant analysis, various metrics have been proposed in the literature for measuring such similarity, based on serial features extracted from the original time series [95,96,97,98], parameters from models [99,100,101,102,103], complexity of the time series [104,105,106,107,108,109], properties of the predictions [110,111] and the comparison of raw data [112]. A description about time series clustering was reported by Vilar and Montero [113].

Furthermore, the sPLS-DA has been applied in a previous study for the classification of different time series of RH [10] using centroid distance. In addition, BER was used to compare the performance of various models. In this research, the variables computed as input for sPLS-DA were the following: estimates of parameters from seasonal ARIMA-GARCH, the last level of smoothing components from SH-W and features extracted from the original time series (ACF, PACF, spectral density, MR and mean). A common seasonal ARIMA-TGARCH was used for the same “stage” of the different time series because it was necessary to deal with the same number of variables as input for sPLS-DA. For the SH-W method, the last level of smoothing components was used instead of RH predictions because each series was split into different stages, according to the seasons and structural breaks identified per season. Regarding the results of the aforementioned study, sPLS-DA with ARIMA-GARCH yielded the best results according to BER. The second best results were achieved using sPLS-DA with SH-W. With respect to sPLS-DA carried out with variables extracted from the original time series, the results of the aforementioned work were compared with previous studies using PCA directly applied to the time series [9].

In the present study, considering that SH-W is intended for producing point forecasts [49] and that two time series are similar if their forecasts are alike [111], predictions of T were included as additional variables. However, according to results from the sPLS-DA, these variables did not show up as important for the classification of sensors. As an exception, by applying the RF, one prediction of T was selected among the relevant variables. Furthermore, the first five coefficients of Wold decomposition were computed using the estimates of parameters for the best ARMA model per sensor. Previously, the ARIMA polynomials were determined according to the seasonal ARIMA that best fitted the time series. The ’best model’ was obtained after one regular and one seasonal differentiation. This procedure made it possible to find the ’best model’ and to compute the same number of classification variables for each sensor. However, results from the sPLS-DA and RF were better when using a common seasonal ARIMA, for all sensors in the same stage of the series. By contrast, the previous study [10] used a common ARIMA-GARCH model and did not compute forecasts for SH-W. Expanding the methods in this research, by using the Wold decomposition and predictions of T from SH-W, did not improve the results.

Few parameters of the seasonal ARIMA models and Wold decomposition were found as relevant for the classification of sensors. Regarding the SH-W, most of the important variables were the last level of smoothing components, but, in the other methods, most of the key variables were features extracted from the model residuals, as discussed below. Actually, the percentage of important variables selected from the SH-W that do not correspond to estimates from residuals was 60.0% and 74.0% for the sPLS-DA and RF methods, respectively. Such appropriate results could be explained by the flexibility of this method. The associated weight becomes higher for the most recent observation, which generates both reliable forecasts and smoothing components for a wide range of time series [49]. From sPLS-DA, the percentage of key variables selected that do not correspond to estimates from residuals was 25.0% and 24.0% for the seasonal ARIMA and Wold decomposition, respectively. From RF, these percentages were 33.3% and 21.1%, respectively. The main reason for these low percentages might be the similarity between the time series and the fact that the estimates of parameters were very similar. Given that residuals account for the variability not explained by the models, their information was relevant in this case for classifying the different time series.

According to the BER values from sPLS-DA, M1 performed better than M2. Moreover, the latter yielded better results than using M3, which in turn led to a better classification than with M4. Regarding the results from sPLS-DA, compared with those from the previous study [10], in both cases, the second best results were derived from the SH-W, possibly because it reliably generates the last level of the smoothing components for a wide range of time series. In the previous research, seasonal ARIMA-TGARCH yielded the best result, while in the present work it was achieved using different functions applied to the time series.

With respect to values of the OOB error from the RF, the use of functions applied to different time series (M1) performed better than when applying seasonal ARIMA (M3), which in turn achieved better results than when using SH-W (M2). The worst results were derived from the Wold decomposition (M4). When using all variables from the four methods, two of them were more important, depending on the number of variables selected per method, as follows. For sPLS-DA and RF, the most relevant method was M2, possibly due to the flexibility of the exponential smoothing. The second most important was M1 for sPLS-DA and M3 for RF, although the percentage of variables from M1 was just 3% less than M3. Results from sPLS-DA were consistent with the BER values, calculated separately for each method because, according to BER, M1 and M2 appeared as the most efficient. By contrast, different results were derived from the RF procedure because M1 and M3 appeared as the best methods, according to OOB error.

The most important variables from sPLS-DA and RF are the following. for M1, PACF at lag 2 and the mean values of ACF, which explain the autocorrelation of the different time series; for M2, seasonal Component 18, mean of ACF values that account for the autocorrelation of the residuals from the SH-W method and the residual variance (SSE); for M3, the first term of MA was a key variable, as well as two others explaining the autocorrelation according to the ARIMA model (PACF at lag 2 and mean ACF values) and the residual variance of the model; and for M4, the first coefficient of the Wold decomposition explaining the autocorrelation of T, PACF at lag 5 that accounts for the autocorrelation of residuals from the ARIMA model and the residual variance. In summary, for M1, the key variables for classifying the different sensors explain the dynamic dependency of the time series of T. In contrast, for M2–M4, some of the most relevant variables explain the dynamic dependency of time series and the residuals.

One disadvantage of sPLS-DA is that it is necessary to know a priori the number of clusters of the time series for their classification. Different indices were employed to determine the number of clusters, and the k-means was used to establish the class for each sensor. The results led to a classification of the series which was consistent with the areas of the museum and the knowledge of the microclimate in this site.

For the task of identifying the key variables that characterize each cluster of sensors, the centroid per class was proposed, according to the first two components. Another solution was to use the mean for each variable, in each class, to identify the class where the mean was the highest. This solution helped to provide a value for characterizing each zone, using the main variables.

By using sPLS-DA with estimates from the seasonal ARIMA, SH-W method or functions applied to the time series, one advantage is the capability for classifying time series with very similar characteristics. The best option among the different possible inputs depends on the characteristics of the time series. The procedure was based on a previous study using a similar methodology to classify time series of RH [10]. SH-W also turned out to be the second best approach according to BER, as was found in the present study, while the best method was a hybrid model with seasonal ARIMA and GARCH. In contrast, in this research, the best result of BER was found, using functions applied to the time series (M1). Another advantage is that the centroids per class, the distances between centroids and the projection of sensors onto the first two components might be helpful in order to select a subset of representative sensors.

### 4.6. Methodology to Select a Subset of Sensors for Future Monitoring Experiments in the Museum

One drawback of the data-loggers used in the monitoring experiment is that they required a manual download of data every two months. It could be much more efficient to use wireless sensors that transmit the readings to a server or a cloud. Commercial wireless sensors are able to instantly transmit the recordings of indoor air conditions, such as temperature and humidity, among others. In addition, these devices can alert the user via e-mail and/or SMS when the recorded values are outside the range of values established as appropriate. Such limits are defined by users according to either the European standards or the requirements of materials or type of artwork which needs to be preserved. However, wireless sensors are more expensive (about 300–400 euros) than the autonomous type of data-loggers used in the present research (approximate price 30 euros) [28]. Moreover, the former can suffer signal transmission problems in some cases. The number of data-loggers used in the present experiment is too big for the long-term monitoring of indoor conditions in this museum. Certain equilibrium has to be reached between the accuracy required and other factors such as sensor price, maintenance and time required to download the data. Thus, for the long-term monitoring of microclimate conditions, it is necessary to decide the optimal number of sensors. For this purpose, one option is to select a subset of sensors per class, according to the results from sPLS-DA. A methodology is proposed in this research for this purpose. It was assumed that the minimum number of sensors for three clusters should be 15, because three sensors per class are the minimum for applying methodologies such as sPLS-DA, and it is necessary to add a few extra sensors in the case of failure or malfunctioning. Based on this criterion, the recommended position for the representative subset of sensors was decided.

In this study, 27 data-loggers were used because the different zones with similar microclimates were not known a priori, but this amount of sensors seems excessive for long-term monitoring. Employing so many data-loggers on a routine basis is not the best option for monitoring the microclimate in the long term, because it takes a long time to download the data, and nearby sensors offer redundant information. One solution is to make a selection of sensors that capture the relevant information. Thus, it is important to determine how many sensors would be necessary and to establish their location. With the goal of selecting an optimal number of sensors per class and the ‘best’ option among the 27 sensors, the following methodology was applied. The minimum number of training sensors was established as 15, because each class should have at least three sensors in order to calculate the variance of any variable per class or to apply methods such as sPLS-DA. The idea was to select a set of sensors, based on the first two components from sPLS-DA, centroids of the three classes and the distances between each centroid and the position of sensors in the multivariate space (i.e., pair of coordinates using C1 for x-axis and C2 for y-axis for each sensor and the centroid of its class).

The first step consists of deciding the optimal number of sensors per class, which can be computed using the variance of the centroid distances of sensors in the same class. The class with the highest variance should select more sensors, while the opposite applies to the class with lowest variance. Secondly, sensors in the same class were split into three subsets, which were created according to the distances to the centroid. The idea is to draw three concentric circles per class, with three different values of the radius (R1–R3) and the same center (O), the centroid. Each area between circumferences makes it possible to identify the sensors per subset. Thus, there are three groups per class: the first (G1) is determined by the inner circle, the second (G2) by the area between the circumferences with R1 and R2 radius and the third group (G3) by the area between the circumferences with R2 and R3 radius. Thirdly, the optimal number of sensors was selected per subset. Such optimal value can be computed using the variance of the distances of sensors, within the same subset. Finally, in order to guarantee representation for each subset, a sample of sensors per subset was randomly selected, according to M1, which is the method with the lowest value of mean BER values.

In respect to the selection of a subset of sensors using the outcomes from sPLS-DA for M1, the percentages of the sum of variances of the centroid distances of the classes NW, SE and Sk were 35%, 40% and 25%, respectively. A proposal based on the results from sPLS-DA with M1 are the following:☐For NW, the number of representative sensors was 15×0.35=5.25≈6. The 13 sensors in this zone were classified in each of the three concentric circles, as follows: D2, D5, A6 and C6 for the first area (G1); B6, B5, A5 and A4 for the second area (G2); and D6, D3, F, D1 and C5 for the third area (G3). The number of representative sensors selected per group (G1–G3) in this NW class was decided according to the variance of the distances: one sensor in G1, three in G2 and two in G3. The proposed subset of representative sensors is the following: D2 (for G1); B6, A5 and A4 (for G2); and D6 and F (for G3).☐For SE, the number of representative sensors was 15×0.4=6. The eight sensors in this zone were classified according to the concentric circles as: C0 (G1); A and B (G2); and C, C1, B1, D and G (G3). The number of SE sensors selected per group was determined based on the variance of the distances: one sensor in G1, one in G2 and four in G3. The proposed subset of representative sensors is as follows: C0 (for G1); A (for G2); and C, B1, D and G (for G3).☐For Sk, the number of representative sensors was established as 15×0.25=3.75≈4. The six sensors regarded as Sk were classified as: A2 and A3 (G1); B2, B3 and B4 (G2); and C3 (G3). The number of sensors chosen per group was decided according to the variance of the distances: one in G1, one in G3 and two in G2. The proposed subset of representative sensors was: A3 (for G1); B2 and B4 (for G2); and C3 (for G3).

HtA and Tr appeared as the most relevant stages for the sensor discrimination. This result is consistent with a previous research [5], which reported a pronounced temperature gradient at the museum, particularly in summer, caused by the greenhouse effect of the skylight. The identification of the key stages of the time series for discriminating the sensors might help to select and enhance the criteria of adequate sampling intervals in automated systems for microclimate monitoring.

Based on the results, the methodology proposed seems effective for characterizing the indoor air conditions in a heritage building, aimed at preventive conservation. This approach can use variables previously calculated by means of either functions applied to different time series or fitting the SH-W method. Furthermore, sPLS-DA might be useful to select a subset of representative sensors, in order to decide the best location for autonomous data-loggers or wireless sensors.

## 5. Conclusions

The temperature in the L’Almoina museum varies according to the location inside the museum and along the year. In order to define a plan for the long-term monitoring for preserving the artifacts, a statistical methodology was proposed. Some of the most important results found in this research are the following.

Both the sPLS-DA and RF methods were useful for identifying the most important variables that explain the differences among the three zones in the museum. For M1, the relevant variables explain the dynamic dependency of the time series. By contrast, with the other methods, the most important variables explain the dynamic dependency of both, the time series of T and the residuals from either SH-W or ARIMA, although most key variables were computed from residuals. Both approaches showed a good capability for discriminating time series. It was possible to obtain parsimonious models with a small subset of variables leading to satisfactory discrimination. Results from sPLS-DA can be easily interpreted. PCA and k-means with sPLS-DA and RF were effective in establishing the different zones in the site and to discriminate the microclimate of these areas. Furthermore, the stages HtA and Tr were the most relevant ones in order to discriminate the different sensors.

The best method for determining the input of variables for sPLS-DA depends on the characteristics of the time series. The SH-W approach appeared to be more flexible for modeling the different time series and obtaining low values of the classification error rate. By applying SH-W, the percentage of selected variables that do not correspond to residuals was higher than when using seasonal ARIMA and Wold decomposition.

For establishing the most important variables for each zone, the centroids of the two first components from sPLS-DA were used. Another option was to identify the class where the mean value of the selected variable was the highest. Thus, the variable with the highest mean for a class could characterize that class.

The methodology proposed might be useful for characterizing different zones in a building, according to the values of T and RH. Furthermore, it might be helpful to establish the optimal number of sensors, in order to manage resources and to monitor the microclimate according to the European Standards.

Regarding future studies: (1) Other versions of sparse PLS DA could be considered, such as SPLSDA and SGPLS [77], in order to compare the capability of classifying time series. Another unsupervised method could also be used to establish the classes before applying sPLS-DA. For example, Guha et al. [114] recently proposed a novel Bayesian-nonparametric strategy for setting the number of clusters and their labels. (2) Further studies about the indoor air conditions at this museum should focus on the time series analysis of both T and RH, in order to improve the characterization of zones with a similar microclimate. (3) The proposed methodology in this paper might be implemented in ’real time’ monitoring so that corrective actions might be adopted in the case of inappropriate measurements. (4) Work in progress is currently applying advanced statistical methods to study the relationship between time series of T and the height gradient in a typical church in a Mediterranean climate.

## Figures and Tables

**Figure 2 sensors-21-04377-f002:**
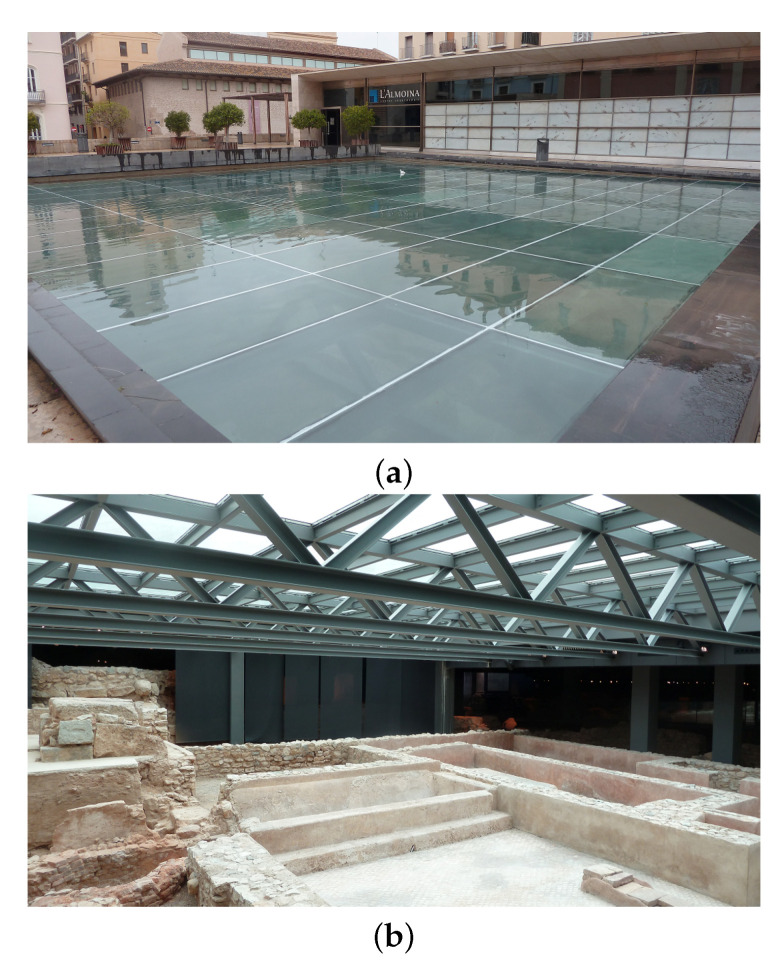
View of the skylight covering part of L’Almoina archaeological site: (**a**) external view from the pedestrian plaza; and (**b**) internal view.

**Figure 3 sensors-21-04377-f003:**
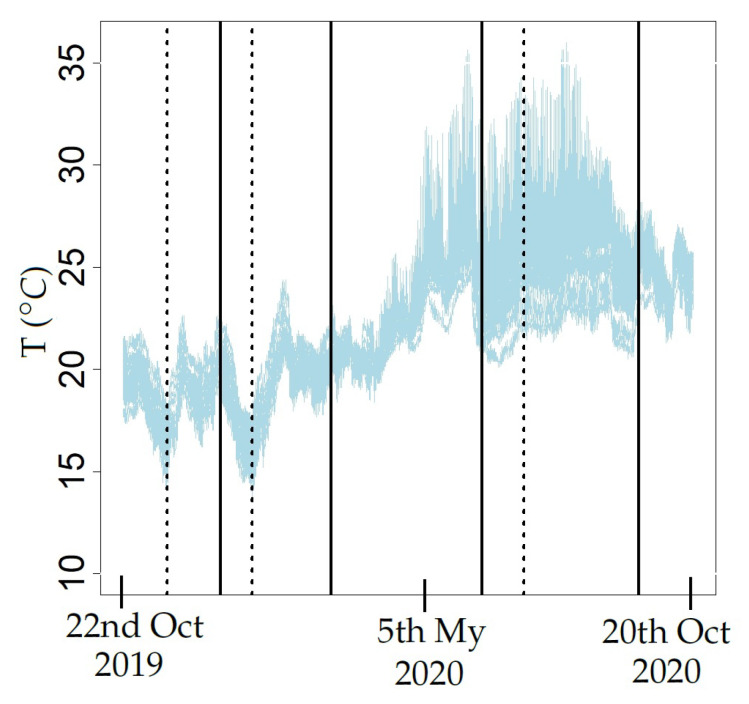
Trajectories of the different time series of T from the 27 sensors located in the museum. Values were recorded between 22 October 2019 and 20 October 2020. The separation of different stages (Wr1, Cd, Tr, Ht and Wr2) is indicated by means of solid vertical lines. Dashed vertical lines indicate the structural breaks identified within the stages Wr1, Cd and Ht.

**Figure 4 sensors-21-04377-f004:**
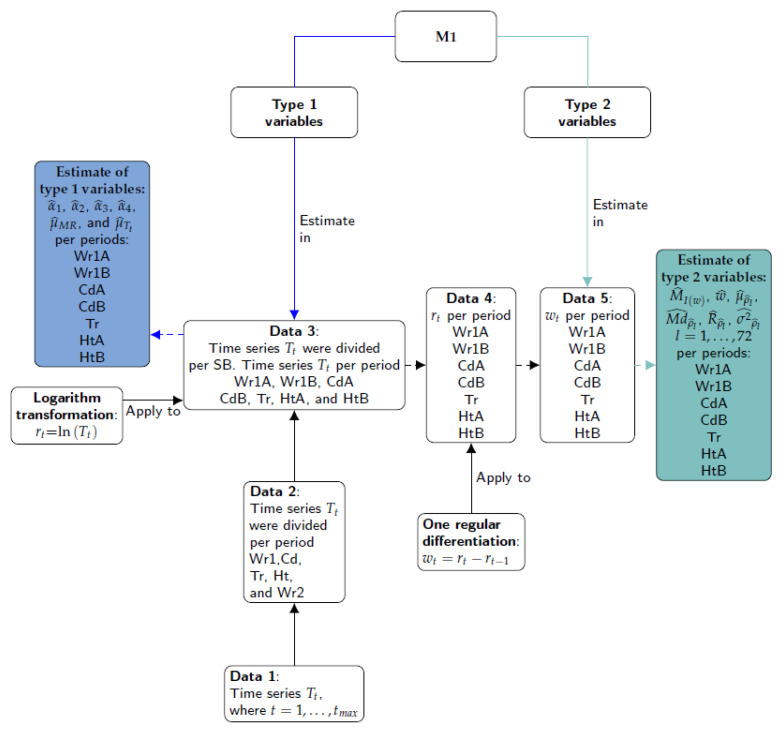
Summary of steps involved in method M1: blue lines, Type 1 classification variables; green lines, Type 2 variables; solid lines, process; dashed line, results. Different boxes contain the name of the stages (i.e., Wr1A, Wr1B, CdA, CdB, Tr, HtA, HtB and Wr2) to indicate that the procedure was applied to all parts of the time series. A structural break was found in Wr1, Cd and Ht, so that the suffixes A and B denote the substages before and after the break, respectively.

**Figure 5 sensors-21-04377-f005:**
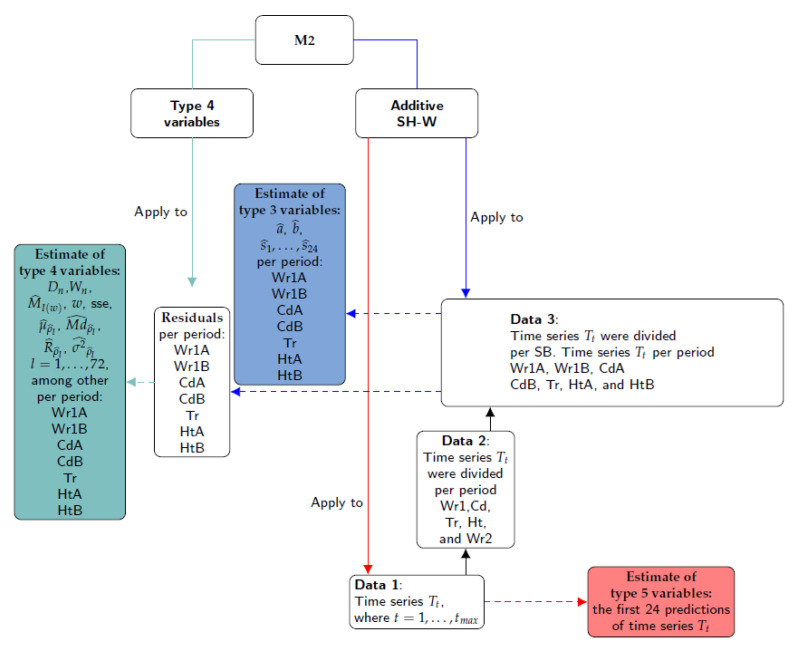
Summary of steps in method M2: blue lines, Type 3 classification variables; green lines, Type 4 variables; red lines, Type 5 variables; solid lines, process; dashed line, results.

**Figure 6 sensors-21-04377-f006:**
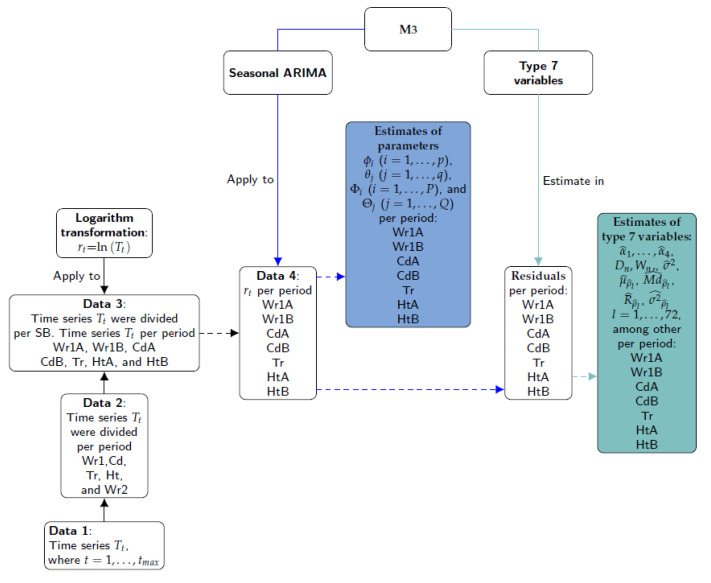
Summary of steps in method M3: blue lines, Type 6 classification variables; green lines, Type 7 variables; solid lines, process; dashed line, results.

**Figure 7 sensors-21-04377-f007:**
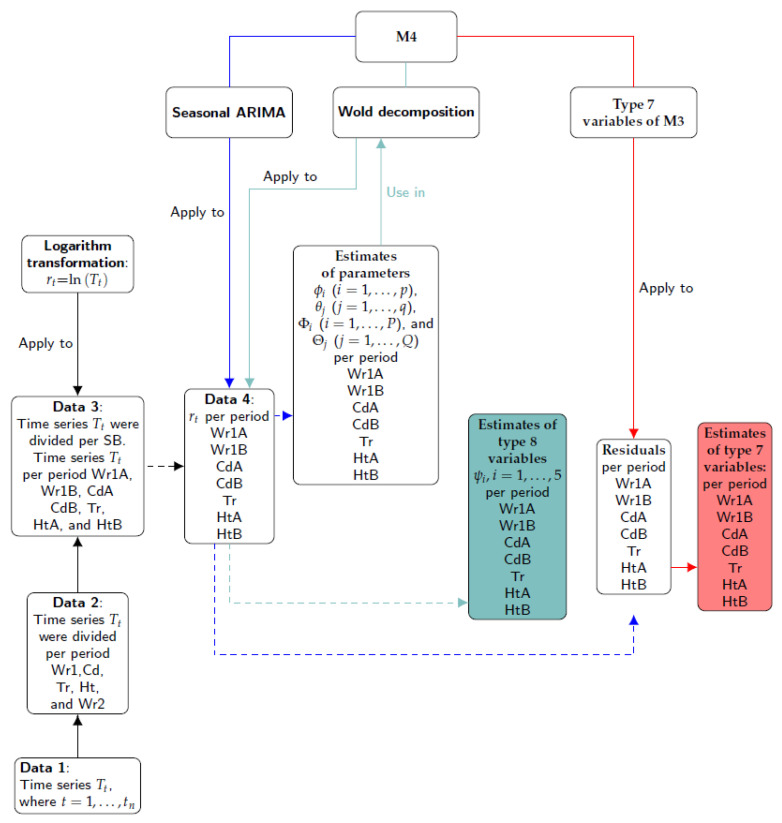
Summary of steps of method M4: red lines, Type 7 variables of M3; green lines, Type 8 variables; solid lines, process; dashed line, results.

**Figure 8 sensors-21-04377-f008:**
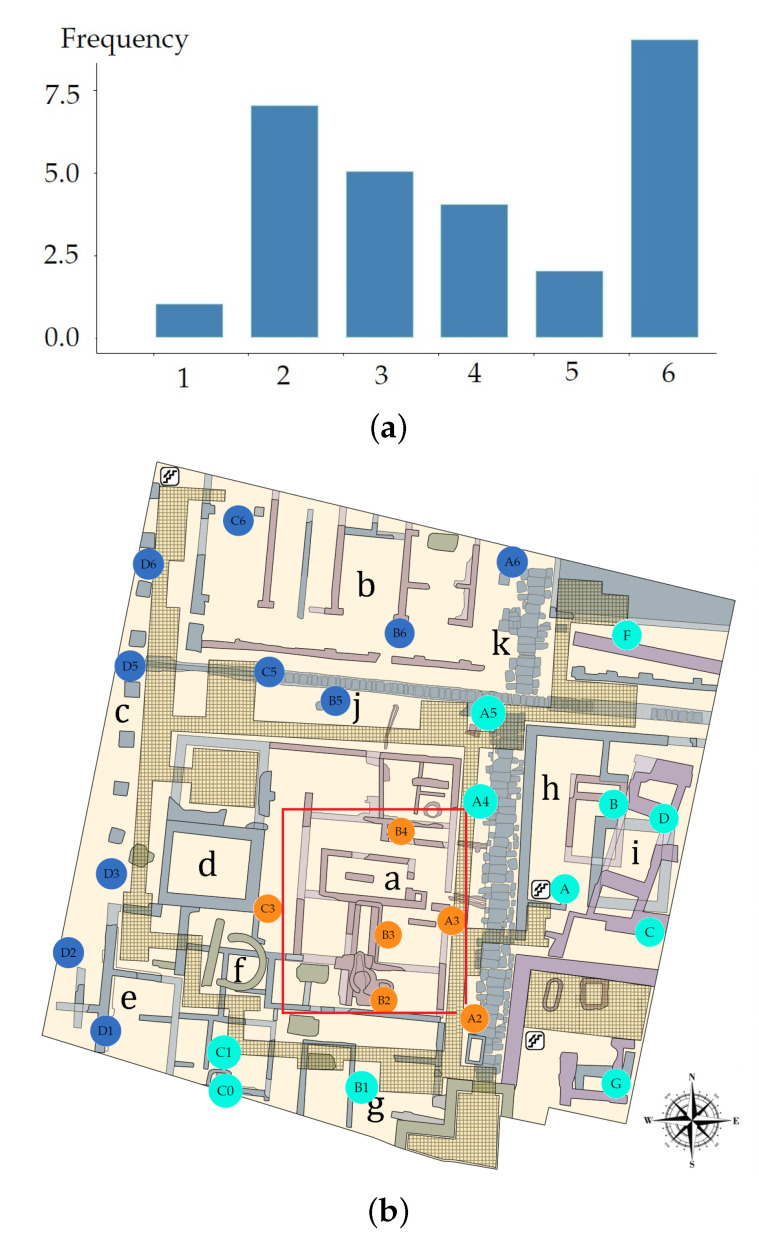
Results associated with the k-means method. (**a**) Absolute frequency (number) of indices that indicates the best number of classes in the museum. For example, two classes are selected by seven indices. (**b**) Classification of sensors installed in the museum, according to the k-means method. Each color (blue, green and orange) corresponds to a different class.

**Figure 9 sensors-21-04377-f009:**
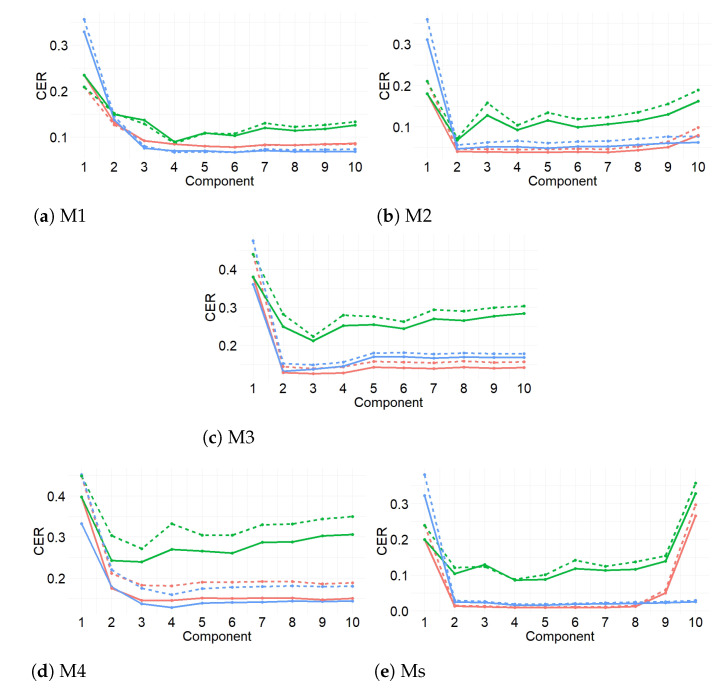
Evaluation of the PLS-DA performance for the classification of sensors into three categories. Vertical axes indicate the classification error rate (CER) for each prediction distance as a function of the number of components (horizontal axis) for: M1 (**a**); M2 (**b**); M3 (**c**); M4 (**d**); and using the variables from all methods (**e**). Three types of prediction distances were considered: Mahalanobis (green lines), maximum (blue lines) and centroid (red lines). Two types of CER were computed: balanced error rate (dashed lines) and overall error rate (solid lines). PLS-DA was carried out using repeated three-fold CV 1000 times.

**Figure 10 sensors-21-04377-f010:**
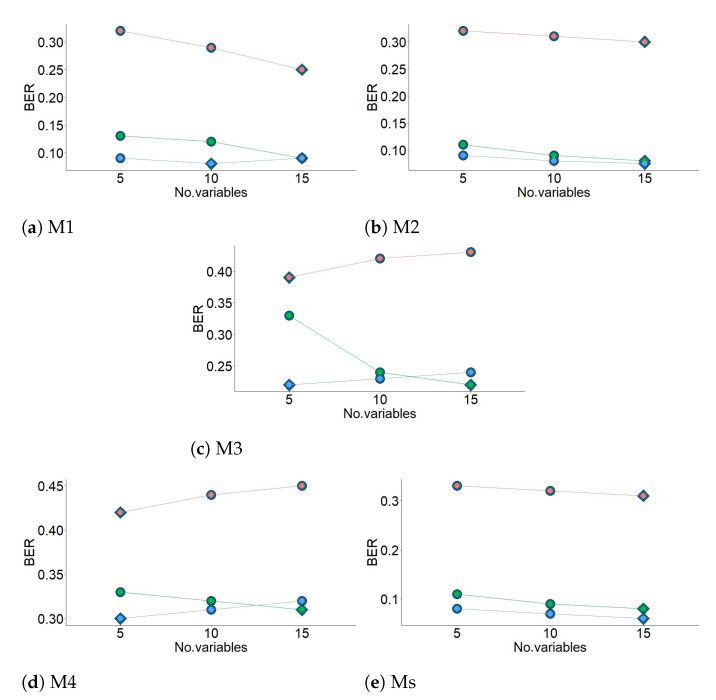
Evaluation of the PLS-DA performance (considering three components) for the classification of sensors into three categories. Vertical axes indicate the Balance error rate (BER) per component (orange lines, Component 1; green lines, Component 2; and blue lines, Component 3). BER values were computed across all folds using 5, 10 or 15 variable (horizontal axes) for eachmethod: M1 (**a**);M2 (**b**);M3 (**c**);M4 (**d**); and all methods (Ms) (**e**). The three-fold CV technique was run 1000 times, using maximum distance prediction. Diamonds highlight the optimal number of variables per component.

**Figure 11 sensors-21-04377-f011:**
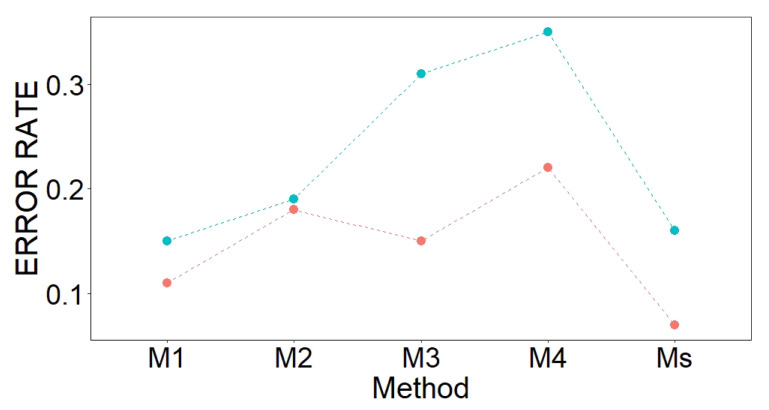
Error rates derived from the sPLS-DA and RF algorithms. Red points are OOB classification error rates per method, based on different sets of classification variables: M1, M2, M3, M4 and all variables (Ms). Blue points are mean values of BER for all the components, per method, for sPLS-DA.

**Figure 12 sensors-21-04377-f012:**
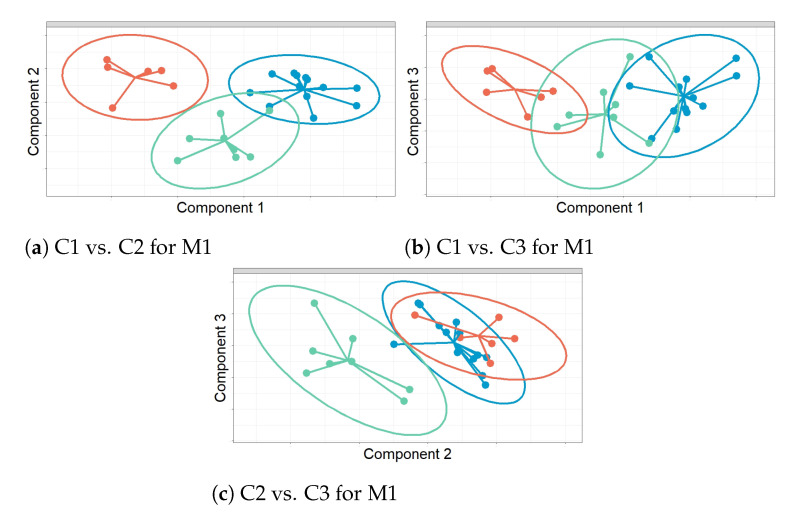
Projection of sensors over the three relevant components (C1–C3) from sPLS-DA, per method (M1–M4) or when using all variables (Ms). Graphs show discrimination of the sensors, according to three classes: North Western (NW), South Eastern (SE) and Skylight (Sk). Color codes: NW sensors in blue, Sk in orange and SE in green. Each graph displays a confidence ellipse for each class (at a confidence level of 95%), in order to highlight the strength of the discrimination.

**Figure 13 sensors-21-04377-f013:**
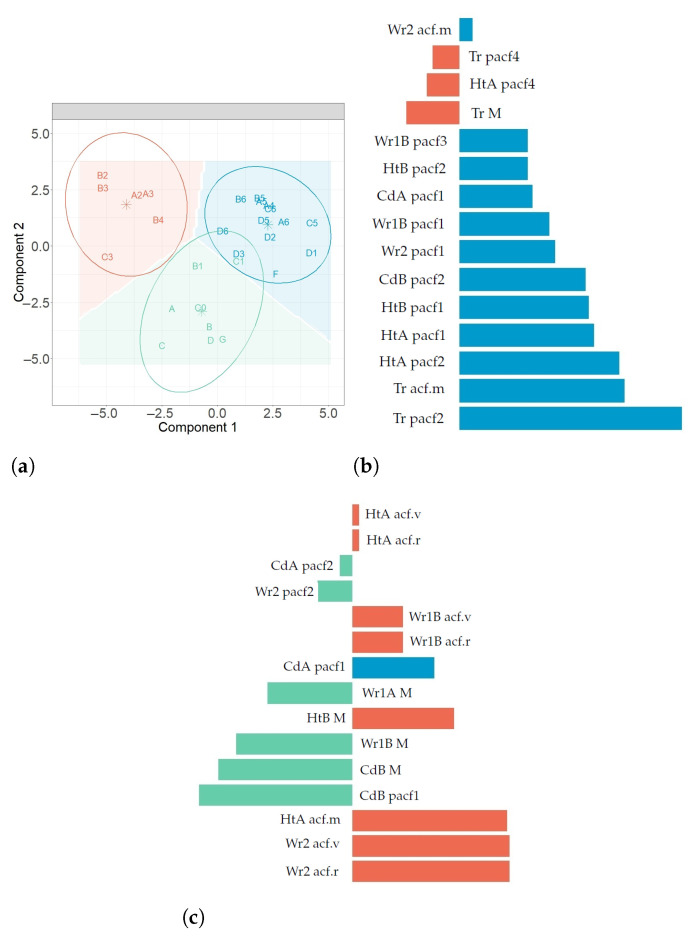
(**a**) Projection of sensors over components C1 and C2 from sPLS-DA for M1. The graph displays a good discrimination of the sensors according to the classes. They are color coded according to the zone where the sensor is located: NWin blue, SE in green and Sk in orange. The most important variables for (**b**) C1 (**c**) and C2, according to the absolute value of their coefficients, are ordered from bottom to top. The color corresponds to the zone in which the variable yields the highest mean component value.

**Table 1 sensors-21-04377-t001:** The most successful models per stage of the different observed time series r are presented in the second column. Column 3 presents the percentages of the LBQ test on the different lags from nt to nt+48 from the 27 sensors that fulfill the assumptions of independence. Column 4 presents the percentages of the LM test from the 27 sensors that fulfill the assumptions of the absence of Arch effect. The significance level used was 0.01.

Stage	Model	LBQ	LM
**(a)** For Method 3			
Wr1A	Seasonal ARIMA$\left(0,1,2\right){\left(2,1,0\right)}_{24}$	77.00	92.52
Wr1B	Seasonal ARIMA$\left(0,1,0\right){\left(2,1,0\right)}_{24}$	3.00	44.44
CdA	Seasonal ARIMA$\left(0,1,0\right){\left(2,1,0\right)}_{24}$	25.00	77.77
CdB	Seasonal ARIMA$\left(0,1,2\right){\left(2,1,0\right)}_{24}$	18.00	18.52
Tr	Seasonal ARIMA$\left(0,1,3\right){\left(2,1,0\right)}_{24}$	11.00	3.70
HtA	Seasonal ARIMA$\left(1,1,3\right){\left(0,1,1\right)}_{24}$	22.00	22.22
HtB	Seasonal ARIMA$\left(0,1,3\right){\left(2,1,0\right)}_{24}$	0.00	37.04
Wr2	Seasonal ARIMA$\left(0,1,2\right){\left(2,1,0\right)}_{24}$	18.00	37.04
**(b)** For Method 4			
Wr1A	A seasonal ARIMA per sensor	92.59	96.30
Wr1B	A seasonal ARIMA per sensor	51.85	59.26
CdA	A seasonal ARIMA per sensor	81.48	81.48
CdB	A seasonal ARIMA per sensor	48.15	25.93
Tr	A seasonal ARIMA per sensor	25.93	3.70
HtA	A seasonal ARIMA per sensor	59.26	29.63
HtB	A seasonal ARIMA per sensor	25.93	44.44
Wr2	A seasonal ARIMA per sensor	55.55	37.04

**Table 2 sensors-21-04377-t002:** The most successful seasonal ARIMA $\left(p,d,q\right){\left(P,D,Q\right)}_{S}$ model per sensor for different stages, with S=24 and D=d=1.

		Sensors																										
**Stage**	**Order**	**D**	**B**	**A**	**C**	**G**	**B2**	**B3**	**B4**	**A3**	**A2**	**C3**	**B5**	**C5**	**C6**	**D6**	**D5**	**A4**	**A5**	**B6**	**A6**	**F**	**D1**	**D3**	**C0**	**C1**	**B1**	**D2**
Wr1A	*p*	0	0	0	0	0	1	0	0	0	0	0	0	1	1	1	0	0	0	0	1	1	0	1	0	2	2	0
	*q*	2	2	1	1	1	1	2	1	2	3	1	1	1	2	1	3	2	1	0	0	0	2	1	3	0	0	1
	*P*	2	2	2	2	2	2	2	2	2	2	2	2	2	2	2	2	2	2	2	2	2	2	2	2	2	2	2
	*Q*	0	0	0	0	0	0	0	0	0	0	0	0	0	0	0	0	0	0	0	0	0	0	0	0	0	0	0
Wr1B	*p*	1	1	1	1	1	0	2	1	0	1	0	1	0	2	2	1	2	3	1	1	0	0	1	1	1	1	1
	*q*	1	1	1	1	2	1	0	2	2	2	0	1	1	0	0	1	1	0	1	2	2	1	1	1	2	2	2
	*P*	2	2	2	2	2	0	2	2	2	2	0	2	2	2	2	2	2	2	2	2	2	2	2	2	2	2	2
	*Q*	0	0	0	0	0	1	0	0	0	0	1	0	0	0	0	0	0	0	0	0	0	0	0	0	0	0	0
CdA	*p*	1	0	0	0	1	2	0	2	1	1	2	2	1	0	0	1	3	0	0	2	3	0	1	0	1	0	1
	*q*	1	1	2	2	3	0	2	0	1	0	1	1	0	1	0	1	0	2	0	2	0	3	1	1	0	3	1
	*P*	2	2	2	2	0	2	0	2	2	2	0	2	0	2	0	2	0	2	2	0	2	0	1	2	2	2	2
	*Q*	0	0	1	1	1	1	1	1	0	0	1	0	2	2	1	0	2	0	0	1	0	2	2	1	2	0	1
CdB	*p*	0	0	0	0	3	0	1	1	1	1	0	1	0	2	0	0	1	0	0	0	1	3	3	1	3	1	3
	*q*	2	1	3	3	0	3	2	2	1	2	1	2	1	0	1	1	2	3	3	1	0	0	0	0	0	2	0
	*P*	2	2	2	2	2	2	0	0	2	2	2	2	2	2	1	2	2	2	2	2	2	2	2	2	2	2	2
	*Q*	0	0	0	0	0	0	1	1	0	0	1	0	0	0	0	0	0	0	0	0	0	0	0	0	0	0	0
HtA	*p*	2	1	2	1	3	0	0	3	0	2	3	2	0	1	0	0	0	0	1	1	1	0	0	2	0	3	1
	*q*	2	3	0	1	1	1	3	0	4	2	0	1	2	1	3	3	1	1	1	0	1	1	0	2	2	1	1
	*P*	0	0	2	2	0	2	1	2	0	0	2	2	2	2	2	2	2	2	2	2	0	2	0	0	2	0	2
	*Q*	1	1	1	1	1	2	0	0	1	1	0	0	0	0	0	0	0	0	0	2	1	0	1	1	1	1	0
HtB	*p*	2	3	0	0	1	1	3	0	0	0	1	1	0	3	0	0	0	1	0	0	1	0	2	0	2	0	0
	*q*	1	0	2	1	1	2	1	2	0	2	2	1	1	0	3	0	2	2	2	1	0	1	0	3	0	2	1
	*P*	0	2	2	2	2	0	0	2	2	2	2	2	2	2	2	2	2	2	2	2	2	2	2	2	2	2	2
	*Q*	1	0	0	0	0	1	1	0	0	0	0	0	0	0	0	0	0	0	0	0	0	0	0	0	0	0	0
Tr	*p*	2	3	0	0	3	0	1	0	1	0	3	1	3	3	3	3	1	3	0	3	0	3	1	0	3	0	1
	*q*	0	0	1	3	0	1	3	3	1	2	0	2	0	0	0	0	2	0	0	0	3	0	1	3	0	3	2
	*P*	2	2	2	2	2	0	0	2	2	2	2	2	2	2	2	2	2	2	2	2	2	2	2	2	2	2	2
	*Q*	0	0	0	0	0	2	1	0	0	0	0	0	0	0	0	0	0	0	0	0	0	0	0	0	0	0	0
Wr2	*p*	3	2	0	0	0	0	0	2	0	2	0	1	3	2	0	0	2	1	0	3	0	0	0	0	0	0	0
	*q*	0	1	3	0	2	3	3	1	3	0	2	2	0	1	3	3	0	2	2	0	0	3	2	2	3	3	2
	*P*	0	2	2	2	2	2	2	0	2	2	1	2	2	2	2	2	2	2	2	2	2	2	2	2	2	2	2
	*Q*	1	0	0	0	0	0	0	1	0	0	2	0	0	0	0	0	0	0	0	0	0	0	0	0	0	0	0

**Table 5 sensors-21-04377-t005:** Percentages of selected variables per stage of the time series for each component (C) and each method (M). Values were computed according to the information contained in Table 3 and Table 4 (e.g., the value 60.0% for C2 of M2 means that 9 out of the 15 selected variables correspond to HtA, according to Table 3 (b)). The two highest values of each column are highlighted in bold and blue, but only one is selected in case of a percentage > 50%.

	M1			M2		M3		M4			Ms		
**Stage**	**C1**	**C2**	**C3**	**C1**	**C2**	**C1**	**C2**	**C1**	**C2**	**C3**	**C1**	**C2**	**C3**
Wr1A	0.00	6.70	**30.80**	0.00	6.70	0.00	6.70	20.00	0.00	0.00	0.00	0.00	0.00
Wr1B	13.30	**20.00**	7.70	6.70	0.00	0.00	0.00	0.00	6.70	20.00	0.00	0.00	33.30
CdA	6.70	13.30	**23.10**	0.00	26.70	0.00	**26.70**	0.00	13.30	0.00	0.00	20.00	6.70
CdB	6.70	13.30	7.70	0.00	0.00	0.00	6.70	0.00	0.00	**40.00**	0.00	0.00	**53.30**
Tr	**26.70**	0.00	7.70	**40.00**	6.70	**60.00**	6.70	20.00	**26.70**	0.00	**60.00**	6.70	0.00
HtA	**20.00**	**20.00**	7.70	6.70	**60.00**	20.00	**26.70**	**60.00**	20.00	0.00	20.00	**60.00**	6.70
HtB	13.30	6.70	7.70	**33.30**	0.00	20.00	13.30	0.00	6.70	0.00	20.00	0.00	0.00
Wr2	13.30	**20.00**	7.70	13.30	0.00	0.00	13.30	0.00	**26.70**	**40.00**	0.00	13.30	0.00

**Table 7 sensors-21-04377-t007:** Results from the random forest algorithm: percentages of selected variables per stage and method (M1–M4) or when using all variables from the four methods (Ms). M2 was the only method which used ’all observations’ and the ’time series split per stage’ for computing the prediction of the time series of T. For the first column (Stage), ’All’ refers to ’all observations of a time series’ and this category is only used for M2. The two largest percentages per method are highlighted in bold and blue.

Stage	M1	M2	M3	M4	Ms
Wr1A	6.80	0.00	11.10	8.80	0.00
Wr1B	11.40	2.40	3.70	10.50	2.70
CdA	11.40	**19.00**	11.10	15.80	13.50
CdB	11.40	0.00	11.10	7.00	8.10
Tr	**15.90**	**19.00**	**25.90**	**17.50**	**21.60**
HtA	**15.90**	**33.30**	**25.90**	**17.50**	**45.90**
HtB	13.60	9.50	11.10	7.00	2.70
Wr2	13.60	14.30	0.00	15.80	5.40
All	0.00	2.40	0.00	0.00	0.00

## Data Availability

Datasets used are available at http://doi.org/10.5281/zenodo.4716389 (accessed on 23 June 2021).

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
