# Peer review of "Multivariate Time Series Analysis of Temperatures in the Archaeological Museum of L’Almoina (Valencia, Spain)"

_sensors, 2021, doi:10.3390/s21134377_

Round 1
Reviewer 1 Report
First of all, I want to congratulate the authors for their efforts done in this manuscript. The authors have presented an interesting paper in which they analyze the temporal variability of sensed parameters in a museum using different statistical tools. In general terms, the paper needs to solve different issues in order to enhance its clarity. Thus, while some sections must be described with more detail to allow a complete understanding of the novelty of this paper, others must be slightly reduced to help the reador focus on the impact of the results.
Regarding the introduction, authors must reshape some parts to facilitate its comprehension. First, a short initial paragraph must be written describing the problem in which this paper is focused, providing data based on other papers or official reports. Then, a second paragraph describing the way in which other authors are facing this problem should be introduced. A third paragraph detailing the usability of statistic for temporal analysis should be added. Most of this information is already in the introduction; it only must be moved to their corresponding paragraph. Then a paragraph about the aim of the paper focusing on the novelty of their data/methodology and the expected results or hypothesis has to be written. Finally, a short paragraph outlining the structure of the rest of the paper has to finish the introduction.
A new section in which authors analyze state of the art should be included between the introduction and the materials and methods. Most references for this section are already included in different parts of the paper authors only need to reshape it.
Concerning the materials and methods, the authors include some methodological issues in the results, and they have to consider moving this content to the most suitable section.
The results are extremely long in the current manuscript, and the discussion is relatively short compared with the results. Thus I recommend two options. The first option to join both sections. The second option, reconsider moving part of the content of the results to the discussion (for example, 3.9 can be entirely moved to discussion). For example, results should focus on the data description without detailing their impact, analysis, or comparison among methods. It is better to explain all these issues in the discussion. If the second option is preferred, I suggest splitting the discussion into different subsection dealing with different issues.
At the end of the conclusions, the authors must include the future work linked to their results.
Author Response
Reviewer 1
Dear reviewer, thank you very much for your inspiring review which I am sure will greatly improve the readability of our research.
The changes have been highlighted in blue in the new version of the paper.
We have made some grammatical improvement that we have highlighted in orange.
First of all, I want to congratulate the authors for their efforts done in this manuscript. The authors have presented an interesting paper in which they analyze the temporal variability of sensed parameters in a museum using different statistical tools.
In general terms, the paper needs to solve different issues in order to enhance its clarity. Thus, while some sections must be described with more detail to allow a complete understanding of the novelty of this paper, others must be slightly reduced to help the reader focus on the impact of the results.
Regarding the introduction, authors must reshape some parts to facilitate its comprehension.
-
First, a short initial paragraph must be written describing the problem in which this paper is focused, providing data based on other papers or official reports.
We appreciate this comment that will substantially improve the understanding of the work. The requested paragraph was added from line 27 to 54.
-
Then, a second paragraph describing the way in which other authors are facing this problem should be introduced.
The requested paragraph was added from line 55 to 69.
-
A third paragraph detailing the usability of statistic for temporal analysis should be added.
The requested paragraph was added from line 70 to 76.
Most of this information is already in the introduction; it only must be moved to their corresponding paragraph.
-
Then a paragraph about the aim of the paper focusing on the novelty of their data/methodology and the expected results or hypothesis has to be written.
The requested paragraph was added from line 77 to 90.
-
Finally, a short paragraph outlining the structure of the rest of the paper has to finish the introduction.
The requested paragraph was added from line 91 to 95.
-
A new section in which authors analyze state of the art should be included between the introduction and the materials and methods. Most references for this section are already included in different parts of the paper authors only need to reshape it.
A new section was added. This was called Background and added from line 96 to 139. This has the following subsections:
2.1. Studies for the long-term preservation,
2.2. European Standards,
2.3. Characteristics of the L’Almoina museum.
-
Concerning the materials and methods, the authors include some methodological issues in the results, and they have to consider moving this content to the most suitable section.
All “Statistical Methods” were moved to Materials and methods section lines168-568
-
The results are extremely long in the current manuscript, and the discussion is relatively short compared with the results. Thus, I recommend two options.
The first option to join both sections. The second option, reconsider moving part of the content of the results to the discussion (for example, 3.9 can be entirely moved to discussion). For example, results should focus on the data description without detailing their impact, analysis, or comparison among methods. It is better to explain all these issues in the discussion. If the second option is preferred, I suggest splitting the discussion into different subsection dealing with different issues.
According to the first suggestion, the results and discussion were joined in a unique section.
Due to this was too long this was divided by topic in different subsections.
We think that this can improve the presentation of the results and discussion.
The subsections are:
5.1. Identification of Structural Breaks in the Time Series,
5.2. Calculation of Classification Variables - Methods M1, M2, M3, and M4,
5.3. Determination of Number of Classes and Class per Sensor Using PCA and K-means Algorithm,
5.4. Sensor Classification using sPLS-DA,
5.5. Sensor Classification using Random Forest Algorithm,
5.6. Methodology to Select a Subset of Sensors for Future Monitoring Experiments in the Museum.
-
At the end of the conclusions, the authors must include the future work linked to their results.
The paragraph referring to future work that was located in the discussion section of the previous version of paper was moved to the last paragraph in conclusion section of the new version It is located now on line 969 to 978. Additionally, the following idea was added in future studies: (4) We are currently applying advanced statistical methods to study the relation between time series of T and the height gradient of sensors in a typical church in a Mediterranean climate.
Reviewer 2 Report
- In Figure 2(a), the green dots are not so green. In Figure 2(b), the unit for temperature in the vertical axis and the date in the horizontal axis should be revised.
- The details about the sensor calibration including sections 2.2. Data Calibration and 3.1 Data Calibration Results are not necessary, since these are not the focus of the study and further the accuracy of the sensor with ±1 oC error is kind of low.
- Section 3 is a mixture of methods and results, which make it hard to read. I suggest the author separate the methods and results.
- The discussions and the conclusions are too long, please be concise.
Author Response
Reviewer 2
Dear reviewer, thank you very much for your inspiring review which I am sure will greatly improve the readability of our research.
The changes have been highlighted in blue in the new version of the paper.
We have made some grammatical improvement that we have highlighted in orange.
-
In Figure 2(a), the green dots are not so green. In Figure 2(b), the unit for temperature in the vertical axis and the date in the horizontal axis should be revised.
The green color of the circles in Figures 1 a) (old Figure 2(a)) and 7 b) were changed to a lighter green. In Figure 1 b) old (Figure 2(b)) the legends of both axes were changed. In the X axis, the year was specified and it was corrected in an error of 20 by 22 October. On the Y axis, the temperature unit writing was improved.
-
The details about the sensor calibration including sections 2.2. Data Calibration and 3.1 Data Calibration Results are not necessary, since these are not the focus of the study and further the accuracy of the sensor with ±1 oC error is kind of low.
The 3.1 Data Calibration Results section was deleted.
-
Section 3 is a mixture of methods and results, which make it hard to read. I suggest the author separate the methods and results.
All “Statistical Methods” were moved to Materials and methods section lines168-568
-
The discussions and the conclusions are too long, please be concise.
Results and discussion have been merged, These can be verified in lines from 771 to 854
Conclusions were summarized. These can be verified in lines from 943 to 979.